# Genetic association of *ANRIL* with susceptibility to Ischemic stroke: A comprehensive meta-analysis

Na Bai[1☉], Wei Liu[2,3☉], Tao Xiang[1], Qiang Zhou[1], Jun Pu[4], Jing Zhao[3], Danyang Luo[5], Xindong Liu[5], Hua Liu[1]*

1 Department of Neurology, The Third People's Hospital of Chengdu & The Affiliated Hospital of Southwest Jiaotong University, Chengdu, Sichuan, China, 2 Institute of Neuroscience, Kunming Medical University, Kunming, Yunnan, China, 3 Department of Neurology, Nanbu People's Hospital, Nanbu, Sichuan, China, 4 Department of Neurosurgery, The Second Affiliated Hospital of Kunming Medical University, Kunming, Yunnan, China, 5 Nuclear Industry 416 Hospital & The Second Affiliated Hospital of Chengdu Medical College, Chengdu, Sichuan, China

☉ These authors contributed equally to this work.
* liuhua@swjtu.edu.cn

**Data Availability Statement:** All relevant data are within the paper and its Supporting Information files.

**Funding:** Hua Liu received each award. This study was financially supported by the Grant No. 2019-

## Abstract

### Background

Ischemic stroke (IS) is a complex polygenic disease with a strong genetic background. The relationship between the *ANRIL* (antisense non-coding RNA in the *INK4* locus) in chromosome 9p21 region and IS has been reported across populations worldwide; however, these studies have yielded inconsistent results. The aim of this study is to clarify the types of single-nucleotide polymorphisms on the *ANRIL* locus associated with susceptibility to IS using meta-analysis and comprehensively assess the strength of the association.

### Methods

Relevant studies were identified by comprehensive and systematic literature searches. The quality of each study was assessed using the Newcastle-Ottawa Scale. Allele and genotype frequencies were extracted from each of the included studies. Odds ratios with corresponding 95% confidence intervals of combined analyses were calculated under three genetic models (allele frequency comparison, dominant model, and recessive model) using a random-effects or fixed-effects model. Heterogeneity was tested using the chi-square test based on the Cochran Q statistic and $I^2$ metric, and subgroup analyses and a meta-regression model were used to explore sources of heterogeneity. The correction for multiple testing used the false discovery rate method proposed by Benjamini and Hochberg. The assessment of publication bias employed funnel plots and Egger's test.

### Results

We identified 25 studies (15 SNPs, involving a total of 11,527 cases and 12,216 controls maximum) and performed a meta-analysis. Eight SNPs (rs10757274, rs10757278, rs2383206, rs1333040, rs1333049, rs1537378, rs4977574, and rs1004638) in *ANRIL* were significantly associated with IS risk. Six of these SNPs (rs10757274, rs10757278,

YF05-00014-SN from Chengdu Municipal Bureau of Science and Technology(http://cdst.chengdu. gov.cn/), and Grant No. 19ZD001 from Health Commission of Sichuan province(http://wsjkw.sc. gov.cn/). The funders had no role in study design, data collection and analysis, decision to publish, or preparation of the manuscript.

**Competing interests:** The authors have declared that no competing interests exist.

rs2383206, rs1333040, rs1537378, and rs4977574) had a significant relationship to the large artery atherosclerosis subtype of IS. Two SNPs (rs2383206 and rs4977574) were associated with IS mainly in Asians, and three SNPs (rs10757274, rs1333040, and rs1333049) were associated with susceptibility to IS mainly in Caucasians. Sensitivity analyses confirmed the reliability of the original results. Ethnicity and individual studies may be the main sources of heterogeneity in *ANRIL*.

## Conclusions

Our results suggest that some single-nucleotide polymorphisms on the *ANRIL* locus may be associated with IS risk. Future studies with larger sample numbers are necessary to confirm this result. Additional functional analyses of causal effects of these polymorphisms on IS subtypes are also essential.

## Introduction

Stroke is the second leading cause of death in the world [1] and the first leading cause of death in China [2]. In 2017, the National Epidemiological Survey of Stroke in China (NESS-China) from 31 provinces reported that the incidence and mortality rates of stroke were 246.8 and 114.8 per 100,000 person-years, respectively, and it is estimated that about 3.4 million new stroke cases occur each year [3]. Stroke warrants some of the highest medical costs in China, costing nearly 75.6 billion yuan (RMB) in direct medical costs [4]. Hospitalization expenses are projected to increase significantly with the expected improvement in people's living standards [5]. Ischemic stroke (IS) accounted for 43.7%–78.9% of all stroke cases in China [6]. IS is a complex disorder with a strong genetic component [7]. Thrombosis of brain arteries secondary to atherosclerosis is considered one of the major pathophysiological mechanisms of IS [8]. Thus, studies into genetic susceptibility to atherosclerosis have attracted a lot of attention.

*ANRIL* (antisense non-coding RNA in the *INK4* locus), which belongs to the long non-coding RNA family, was found to have a strong association with the risk for cardio-metabolic diseases [9], playing a key role in atherosclerotic diseases such as IS. A number of studies have explored the relationship between *ANRIL* and IS across populations worldwide. However, most of these studies used small sample sizes and the findings were inconclusive. Data from linkage and association studies showed that susceptible locus for common diseases had only minimal effects. Meta-analysis is a powerful tool that allows the detection and validation of minimal biological effects in human genetic association studies [10]. Researchers have investigated the role of a few single-nucleotide polymorphisms (SNPs) on the *ANRIL* locus in IS across different populations by meta-analysis. However, the association of other genetic variants and other SNPs in *ANRIL* with IS deserves further analyses. In addition, some recently published studies across ethnicities were found in the literature search. In this study, we conducted an updated meta-analysis on all available association study data to comprehensively evaluate the contribution of *ANRIL* to the risk of IS.

## Materials and methods

### Study design

This research was conducted according to the PRISMA (Preferred Reporting Items for Systematic Reviews and Meta-Analysis) statement and the guidelines presented in Systematic

Reviews of Genetic Association Studies by Sagoo *et al.* [10]. *ANRIL* polymorphism was used as the exposure and IS as an outcome. This work did not require the approval of an ethics committee and was not registered in any database. The completed PRISMA checklist and Meta-analysis on Genetic Association Studies Checklist are given in S1, S2 Appendices.

## Data collection

All studies involving the relationship between *ANRIL* gene polymorphisms and stroke were identified independently by three investigators (Bai N, Liu W, and Zhou Q) by searching the following databases until August 2021: PubMed (from 1966), EMBASE (from 1966), the Cochrane Library (from 2003), ProQuest Dissertations & Theses Database (from 1980), Biosis Preview (from 1990), Web of Science (from 1990), China National Knowledge Infrastructure (CNKI, from 194), and Wanfang Database (including journal articles, dissertations or theses, and conferences literature, from 1990). We used the following keywords or their combinations in search strategies: "*ANRIL*", "*CDKN2B-AS1*", "antisense non-coding RNA in the *INK4* locus", or "9p21" and "stroke", "cerebral infarction", or "cerebrovascular disease". We limited the search to only human studies. Examples of the keywords search strategy in PubMed are: ("ANRIL"[All Fields] OR "CDKN2B-AS1"[All Fields] OR "antisense non-coding RNA in the *INK4* locus"[All Fields] OR "9p21"[All Fields]) AND ("stroke"[All Fields] OR "cerebral infarction"[All Fields] OR "cerebrovascular disease"[All Fields]).

The references listed in the retrieved articles and in review articles as well as abstracts from recent conferences were also searched for possible eligible studies. Only the most recent or complete reports were selected for analysis if the same or a similar patient cohort was included in several publications. There were no restrictions on the source of the control group, and studies in which the control groups were not in Hardy-Weinberg equilibrium were excluded [11].

Studies meeting the following criteria were included for meta-analysis: 1) genetic association studies of the *ANRIL* polymorphisms with IS were performed using a population (hospital)-based, case-control, nested case-control, or cohort design; 2) IS was diagnosed using a standard that has been widely accepted; 3) control subjects were unrelated individuals, with no symptomatic vascular disease as confirmed by physicians; 4) genotype or allele frequencies were reported in both patients with IS and in controls or could be calculated successfully; and 5) a genetic variant of *ANRIL* was included in at least two of the studies. Case-only studies, family-based studies, and review articles were excluded. The quality of included studies was assessed based on the published study [12] and the Newcastle-Ottawa Scale (NOS) [13]. A NOS score $\geq 7$ was considered high quality [13].

## Data extraction

Data were carefully extracted from all eligible studies independently by two authors (Liu W, Xiang T), and any disagreements were resolved by discussion. The following information was extracted: first author's surname, year of publication, country of origin, study design, sex composition of the case and control groups, ethnicity of the subjects studied, total number of subjects, definition and characteristics of cases and controls, genetic variants associated with IS, genotyping methods, distribution of genotypes and alleles, IS subtype (if reported), information on additional genetic variants, as well as gene–gene and gene–environment interactions (if investigated). Genotype frequencies were calculated where possible.

For studies that included subjects from different ethnic groups, data were extracted separately for each ethnic group. When some of the information was not available, we contacted the corresponding author by email for additional information.

## Statistical analyses

Odds ratios (ORs) and pooled ORs with corresponding 95% confidence intervals (CIs) were calculated using the fixed-effects or random-effects model. For the chi-square test based on Cochran Q statistic, p-values <0.10 were considered to be statistically significant [14]. The $I^2$ metric was used to evaluate the heterogeneity among studies [15].

Hardy-Weinberg equilibrium was tested in the control groups using the chi-square test. Three genetic models were used to examine the association of *ANRIL* polymorphisms and risk of IS: (1) allele contrast (AC) (effect of each additional risk allele), (2) dominant model (DM), and (3) recessive model (RM). Multiple testing correction was conducted using the false discovery rate (FDR) method proposed by Benjamini and Hochberg. Inverted funnel plots and Egger's test were performed to detect publication bias in the analyses involving different genetic variants. Publication bias was considered to be present if the inverted funnel plot was asymmetric and/or Egger's test result was significant (p <0.10).

Sub-population analyses were conducted for ethnicity [16], and subgroup analyses for IS subtype, age, or sex (if available) were also performed [17]. A sensitivity analysis was performed with the exclusion of specific studies [18], such as poor-quality studies (NOS <7) or studies where no *ANRIL* genetic variants were found in either cases or controls. All statistical analyses were performed with the Cochrane Review Manager (RevMan, version 5.4) and STATA 16.0 package. A probability value of p<0.05 (two-tailed) was considered significant unless indicated otherwise.

## Results

### Study selection and characteristics of eligible datasets

We found 856 records by primary searches in the databases and six additional records were identified from other sources, including 113 articles from English-language databases and 749 items from Chinese-language databases. Initially, 115 potentially relevant articles (16 in Chinese and 99 in English) were initially selected after reading the titles and abstracts. After reading the full text of these articles, 90 articles were excluded because of duplicates, reviews, mixed samples (transient ischemic attack or hemorrhagic stroke were not excluded), insufficient data, irrelevant content, genetic variants beyond the scope of this study, or ineligible study design. Finally, 25 articles (2 in Chinese and 23 in English) [19–43] involving 15 SNPs (rs2383207, rs10757274, rs10757278, rs2383206, rs1333040, rs1333049, rs1537378, rs4977574, rs1004638, rs7865618, rs10965227, rs1333042, rs7044859, rs10116277, and rs10757269) were found to be eligible for the meta-analysis after applying all the inclusion and exclusion criteria described above. The results of the systematic literature search and article selection are summarized in Fig 1. The excluded articles and the reasons for excluding each article are given in S3 Appendix.

Twenty-three of the included articles were full-length reports published in peer-reviewed journals [19–27, 29–33, 35–43], and two were Master degree thesis [28, 34]. The characteristics of these studies and the *ANRIL* polymorphisms involved in the meta-analysis are summarized in Table 1. A summary of the total number of studies on different *ANRIL* SNPs is provided in Table 2.

Most of the included studies had NOS scores of 7–9, four studies had NOS scores of 6 [26, 32, 35, 39], and two studies had NOS scores of 5 [20, 29].

### Genetic association of 15 *ANRIL* SNPs with IS

**SNP rs2383207.** The association of rs2383207 with IS risk was investigated in 12 studies [21, 23, 28–30, 32, 35, 38–41, 43] involving 11, 527 cases and 12, 216 controls.

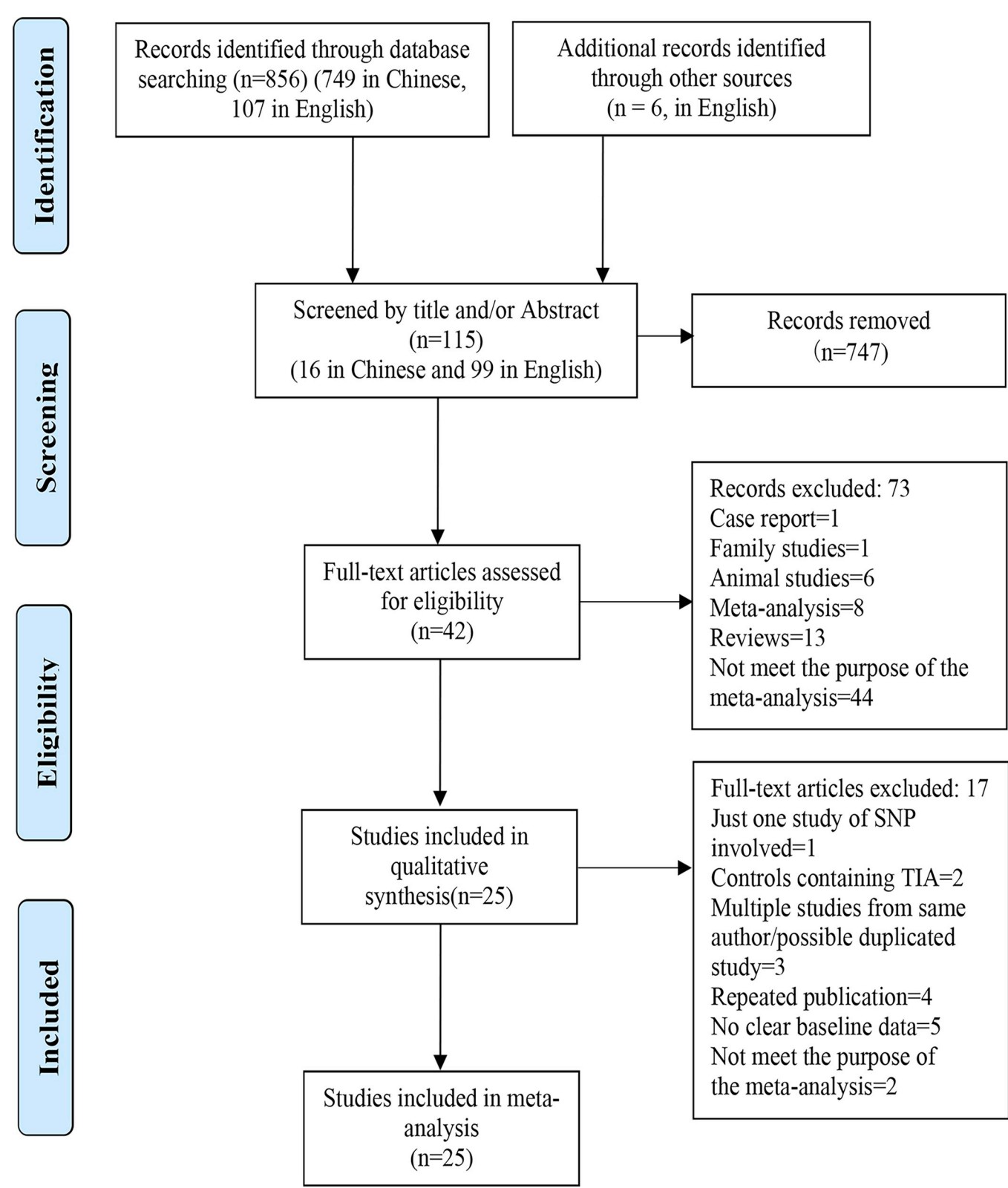

**Fig 1. Flowchart of the literature search and article selection for the meta-analysis.**

**Table 1. Characteristics of included studies and *ANRIL* polymorphisms for meta-analysis.**

| Studies (Year) | Countries Population | Variants | Samples Selection/Characteristics | | NOS Score |
|---|---|---|---|---|---|
| | | | Cases | Controls | |
| Zee RY 2007 [19] | US White Caucasians | rs10757274 rs2383206 | Entire IS. N = 254, age: 61.0±0.3. Men alone. | N = 254, PB, age: 60.8±0.3. Men alone. | 9 |
| Helgadottir A 2008 [20] | Iceland Sweden Caucasians | rs10757278 | IS (LAA and CE). N = 491. No description of age and gender. | N = 14993, PB. No description of age and gender. | 5 |
| Smith JG 2009 [21] | Sweden Caucasians | rs2383207 rs10757274 rs1333049 rs1333040 | Entire IS. LSR, N = 1837, age: 73.4±12.0. F: 992 (54%) MDC, N = 888, age: 62.9±6.6. F: 488 (55%). | LSR, N = 947, age: 73.2±11.9. F: 540 (57%); MDC, N = 893, age: 62.9±6.6. F: 482 (54%). | 8 |
| Hu WL 2009 [22] | China Asians | rs10757274 rs2383206 | Entire IS. N = 355, age: 58.72±10.87. F: 95 (26.8%). | N = 430, HB, age: 60.4±10.91. F: 130 (30.2%). | 7 |
| Gschwendtner A 2009 [23] | | rs7044859 rs7865618 rs1537378 rs2383207 rs10757278 | | | 7 |
| | Munich (Germany) Caucasians | | N = 1090, age: 65.4±13.5. F: 418 (38.3%). Data of IS subtypes available. | N = 1244, PB, age: 62.4±10.9. F: 471 (37.9%). | |
| | London (UK) Caucasians | | N = 758, age: 66±13.2. F: 314 (41.4%). Data of IS subtypes available. | N = 872, PB, age: 65.3±8.8. F: 374 (42.9%). | |
| | Baltimore (USA) Mixed Populations | | N = 652, age: 41.1±7.3. F: 301 (46.2%). Data of IS subtypes available. White: 327, Black: 275, Other ethnicity: 50. | N = 718, PB, age: 39±7.1. F: 373 (51.9%). White: 384, Black: 271, other ethnicity: 63. | |
| | Jacksonville (USA) Mixed Populations | | ISGS N = 603, age: 64.6±13.8. F: 347 (57.5%). Data of IS subtypes available. White: 445, Black: 139, other ethnicity: 19. | ISGS N = 435, age: 60.7±14.9. F: 272 (62.5%). White: 314, Black: 106, other ethnicity: 15. | |
| | Boston (USA) Mixed populations | | N = 608, age: 65.2±15.7. F: 274 (45%). Data of IS subtypes available. White: 549, Black: 22, other ethnicity: 37. | N = 519, PB, age: 66.8±9.3. F: 270 (52%). White: 498, Black: 8, other ethnicity: 13. | |
| | Aberdeen (UK) Caucasians | | N = 607, age: 69.6±12.2. F: 273 (45%). Data of IS subtypes available. | N = 517, PB, age: 67.1±9.0. F: 252 (48.7%). | |
| Ding H 2009 [24] | China Asians | rs2383206 rs1004638 rs10757278 | N1 = 558, age: 61.0±9.8. F: 196 (35.2%). N2 = 442, age: 63.8±10.6. F: 143 (32.3%). Entire IS. | N1 = 557, age: 62.3±9.3. F: 211 (37.9%). N2 = 502, age: 62.5±8.7. F: 237 (47.2%). PB/HB. F: 448 (42.3%) | 8 |
| Yamagishi K 2009 [25] | USA Caucasians Africans | rs10757274 | N = 13380 (African-Americans: 3499, Whites: 9881) for rs10757274. IS = 524 (African-Americans: 218, Whites: 306). No description of age and gender. | N = 11528 (African-Americans: 2804, Whites: 8724) for rs10757274. No description of age and gender. | 7 |
| | | rs2383206 | N = 12888 (African-Americans: 3399, Whites: 9489) for rs2383206. IS = 516 (African-Americans: 212, Whites: 304). No description of age and gender. | N = 11078 (African-Americans: 2719, Whites: 8359) for rs2383206. No description of age and gender. Notes: the controls is not in HWE in Whites for rs2383206. | 7 |
| Luke MM 2009 [26] | Austria Caucasians | rs10757274 | Entire IS. N = 562, age: 66.0±14. F: 236 (42%). | N = 815, PB, age: 58.8±8.5. F: 418 (51.3%). | 6 |
| Olsson S 2011 [27] | Sweden Caucasians | rs10965227 rs1333040 rs10757278 rs1537378 | Entire IS. N = 844, age: 56±11. F: 290 (34%). | N = 668, PB, age: 56±10. F: 276 (41%). | 8 |

*(Continued)*

**Table 1.** (Continued)

| Studies (Year) | Countries Population | Variants | Samples Selection/Characteristics | | NOS Score |
|---|---|---|---|---|---|
| | | | Cases | Controls | |
| Yue XY 2011 [28] | China Asians | rs10757274 rs10757278 rs2383206 rs2383207 rs1004638 rs1333049 rs1537378 | Entire IS. N = 769, age: 59.91±13.11. F: 257 (33.4%). | N = 682, PB, age: 59.37±11.53. F: 254 (37.2%) | 8 |
| Lin HF 2011 [29] | Taiwan China Asians | rs1333040 rs2383207 rs1333049 | Entire IS. N = 687, age: 64.4±12.4. F: 249 (36.2%). | N = 1377, PB, age: 55.1±12.4.F: 742 (53.9%). | 5 |
| Zhang WL 2012 [30] | China Asians | rs10757274 rs2383206 rs2383207 rs10757278 | N(LAA) = 724, age: 61.5± 9.1. F: 263 (36.3%) N(SVO) = 466, age: 61.0±8.5.F: 169 (36.3%) | N = 1664, PB, age: 59.8±8.2. F: 689 (41.4%). | 8 |
| Wang C 2012 [31] | China Asians | rs1333040 | Entire IS. N = 286, age: 60.37±7.71. Female alone. | N = 831, PB, age: 57.94±8.75.Female alone. | 8 |
| Heckman MG 2013 [32] | USA Caucasians Africans | rs1333040 rs4977574 rs1333042 rs2383207 | N = 264, age: 72±12. F: 117 (44.32%). For SWISS Caucasians, entire IS. N = 449, age: 71±15. F: 184 (40.98%). For ISGS Caucasians, entire IS. N = 166, age: 61±13. F: 84 (50.60%). For ISGS African American, entire IS. Notes: The data of ISGS was removed from last analyses because of its being from the Gschwendtner's study in rs2383207 and rs1333040 | N = 374, PB, age: 72±11. F: 169 (45.19%). For SWISS Caucasians. N = 334, PB, age: 67±15. F: 165 (49.40%). For ISGS Caucasians. N = 117, PB, age: 59±14. F: 69 (58.97%). For ISGS African American. Notes: The controls are not in HWE in Caucasians, SWISS. | 6 |
| Lovkvist H 2013 [33] | Sweden Caucasians LSR MDC SAHLSIS | rs4977574 | N = 3986, age: 70. F: 1775 (44.5%). LAA, SAA and CE. | N = 2459, PB, age: 68. F: 1069 (43.5%). | 7 |
| Zhang T 2014 [34] | China Asians | rs10757274 | LAA alone. N = 229, age: 59.36±11.15. F: 104 (45.41%). | N = 233, PB, age: 58.88±8.17. F: 113 (48.5%). | 8 |
| Lu Z 2015 [35] | China Asians | rs10757278 rs1333049 rs2383206 rs1537378 rs4977574 rs2383207 | N = 153 (Entire IS without carotid plaque), age: 56.56±7.6. F: 57 (37.25%). | N = 258, PB, age: 56.34±7.85. F: 131 (50.78%). | 6 |
| Bi JJ 2015 [36] | China Asians | rs10757278 rs1537378 | LAA alone. N = 116, age: 53.74±12.32. F: 26 (22.41%). | N = 118, PB, age: 53.52±11.98. F: 33 (27.97%). | 7 |
| Cao XL 2016 [37] | China Asians | rs1333040 rs1333042 rs4977574 | Entire IS. N = 569, age: 62.53±11.92. F: 173 (30.4%). Including LAA and SAA | N = 541, PB/HB, age: 6139±11.41. F: 195 (36%) | 8 |
| Akinyemi R 2017 [38] | Nigeria and Ghana Africans | rs2383207 | Entire IS. N = 429, age: 61.34±12.83. F: 231 (53.85%) | N = 483, PB, age: 60.26±12.56.F: 247 (51.14%). | 7 |
| Yang JL 2018 [39] | China Asians | rs1333049 rs2383207 | Entire IS. N = 550, age: 70.10 ± 8.82. F: 244 (44.4%) | N = 550, HB, age: 69.23 ± 9.68. F: 257 (46.7%). | 6 |

(*Continued*)

**Table 1.** (Continued)

| Studies (Year) | Countries Population | Variants | Samples Selection/Characteristics | | NOS Score |
|---|---|---|---|---|---|
| | | | Cases | Controls | |
| Xiong L 2018 [40] | China Asians | rs10757278 rs1004638 rs1333040 rs1333049 rs1537375 rs1537378 rs2383206 rs2383207 rs7044859 rs7865618 rs10116277 rs10757269 rs10757274 | LAA alone. N = 200, age: 59.12±8.65. F: 77 (38.5%). | N = 205, PB, age: 56.87±7.87. F: 94 (45.85%). | 8 |
| Ferreira LE 2019 [41] | Brazil Caucasians | rs2383207 | LAA alone. N = 195, age: 66.9±11.6. F: 72 (36.9%). | N = 249, PB, age: 61.6±10.7. F: 138 (55.4%). | 8 |
| Han XM 2020 [42] | China Asians | rs10757278 | Entire IS. N = 505, age: 59.9±10.9. F: 180 (35.6%) | N = 652, HB, age: 59.0±11.9. F: 253 (38.8%) | 7 |
| Wang Q 2021 [43] | China Asians | rs2383207 rs4977574 | N = 567, age: 61.72±10.17. F: 203 (35.8%). Including LAA and SVO. | N = 552, HB, age: 61.9±9.52. F: 204 (37%). | 7 |

Notes: NOS: Newcastle-Ottawa Scale; HWE: Hardy-Weinberg equilibrium; F: female; PB: population-based; HB: hospital-based; IS: ischemic stroke; LAA: large-artery atherosclerosis; SVO and SAA: small-vessel occlusion; CE: cardioembolism; SWISS: siblings with ischemic stroke study; ISGS: ischemic stroke genetics study; LSR: Lund Stroke Register; MDC: Malmo Diet and cancer Study; SAHLSIS: Sahlgrenska Academy study on ischemic stroke.

No significant association of rs2383207 with IS was found under three genetic models in whole studied population, sub-populations, and IS subtypes. High heterogeneity was detected in the whole studied population (AC: $I^2 = 82\%$, p <0.001; DM: $I^2 = 71.6\%$, p <0.001; RM: $I^2 = 74.5\%$, p <0.001) and in large-artery atherosclerosis (LAA) subtypes (AC: $I^2 = 85.7\%$, p<0.001; DM: $I^2 = 77.6\%$, p<0.001; RM: $I^2 = 76.9\%$, p<0.001) with all three models; however,

**Table 2.** *ANRIL* SNPs included in the meta-analysis.

| SNPs | Studies (n) | Cases (n) | Controls (n) | Composition of studies n (%) | | | |
|---|---|---|---|---|---|---|---|
| | | | | Caucasians | Asians | African | Mixed populations |
| rs2383207 | 12 | 11,527 | 12,216 | 3(25.0%) | 7(58.3%) | 1(8.3%) | 1(8.3%) |
| rs10757274 | 10 | 7,059 | 18,784 | 4(40.0%) | 5(500.%) | 0 | 1(10.0%) |
| rs10757278 | 10 | 9352 | 24552 | 2(20.0%) | 7(70.0%) | 0 | 1(10.0%) |
| rs2383206 | 9 | 4,431 | 8,423 | 2(22.0%) | 6(67.0%) | 1(11%) | 0 |
| rs1333040 | 9 | 6,581 | 8,379 | 4(44.0%) | 5(56.0%) | 0 | 0 |
| rs1333049 | 7 | 5,351 | 6,061 | 2(29.0%) | 5(71.0%) | 0 | 0 |
| rs1537378 | 6 | 6,166 | 6,129 | 1(16.0%) | 4(67.0%) | 0 | 1(16.0%) |
| rs4977574 | 5 | 6,083 | 4,593 | 1(20.0%) | 3(60.0%) | 0 | 1(20.0%) |
| rs1004638 | 3 | 1,959 | 1,941 | 0(0.0%) | 3(100.0%) | 0 | 0 |
| rs7865618 | 2 | 4,303 | 4,477 | 1(50.0%) | 0 | 0 | 1(50.0%) |
| rs10965227 | 2 | 1,395 | 1,223 | 1(50.0%) | 1(50.0%) | 0 | 0 |
| rs1333042 | 2 | 1,281 | 1,220 | 1(50.0%) | 1(50.0%) | 0 | 0 |
| rs7044859 | 2 | 4,322 | 4,461 | 1(50.0%) | 0 | 0 | 1(50.0%) |
| rs10116277 | 2 | 512 | 1371 | 1(50.0%) | 1(50.0%) | 0 | 0 |
| rs10757269 | 2 | 754 | 752 | 0(0.0%) | 2(100.0%) | 0 | 0 |

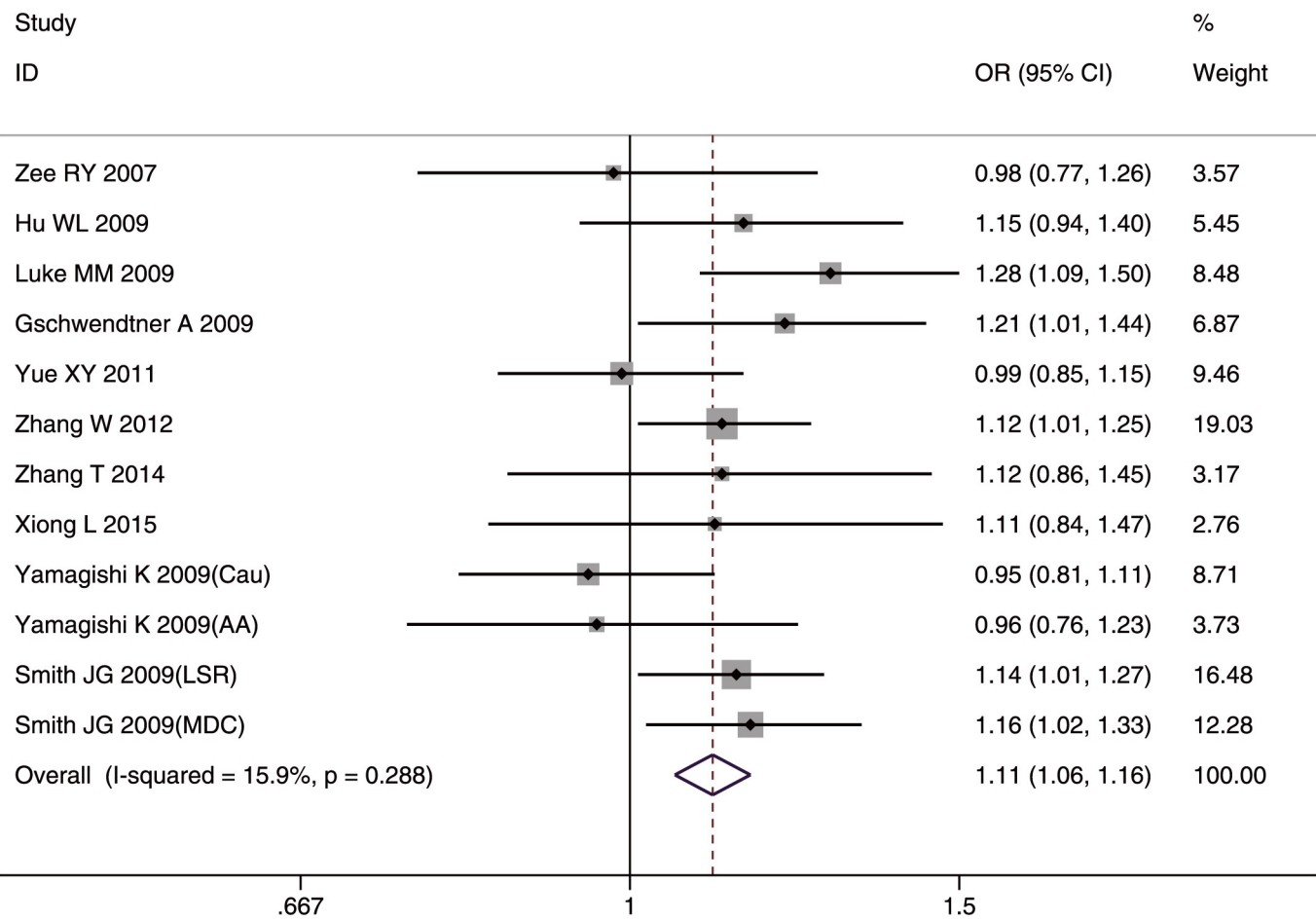

**Fig 2. Forest plot of rs10757274 allele frequency (G vs. A) associated with IS in the whole studied population.**

the heterogeneity disappeared when the Caucasian studies were excluded, suggesting that ethnicity (Caucasian) may be the source of heterogeneity. Meta-regression analysis to identify different sources of heterogeneity indicated that ethnicity may be linked to heterogeneity (p = 0.085), but this finding had no statistical significance.

The sensitivity analysis excluding the poor-quality studies [29, 32, 35, 39] gave similar overall results, confirming that the results were stable and reliable. We did not find publication bias for this SNP using the funnel plots and Egger's test (p = 0.167 in the allelic comparison model).

**SNP rs10757274.** Ten articles [19, 21–23, 25, 26, 28, 30, 34, 40] explored the relationship of SNP rs10757274 (7,059 cases and 18,784 controls) to IS. The G allele was found to have a significant relationship to IS risk in the whole studied population (OR = 1.11, 95%CI: 1.06–1.16, FDR-corrected p (p-FDR) <0.001) (Fig 2) and in the Caucasian studies (OR = 1.13, 95% CI: 1.06–1.20, p-FDR<0.001). The AA genotype conferred a protective effect in the whole studied population (OR = 0.90, 95%CI: 0.83–0.98, p-FDR = 0.0255).

In the IS subtype analyses, the G allele and GG genotype conferred susceptibility to LAA in the whole studied population (G allele: OR = 1.18, 95%CI: 1.08–1.30, p-FDR <0.001; GG genotype: OR = 1.31, 95%CI: 1.13–1.52, p-FDR <0.001), but mainly in Asians (G allele: OR = 1.18, 95%CI: 1.06–1.31, p-FDR = 0.003; GG genotype: OR = 1.33, 95%CI: 1.12–1.57, p-FDR = 0.003). In contrast, the AA genotype had a protective role in LAA only in the whole studied population (OR = 0.84, 95%CI = 0.73–0.96, p-FDR = 0.014). Sex had no effect in any of the comparisons.

Significant heterogeneity among studies was detected only in the recessive model (GG/(AA +AG)) in the whole studied population ($I^2$ = 54.8%, p = 0.018) and in the Caucasians studies ($I^2$ = 78.3%, p = 0.003). The heterogeneity disappeared in the whole studied population ($I^2$ = 40%, p = 0.11) and in Caucasians ($I^2$ = 47%, p = 0.15) after excluding the study by Yamagishi *et al.* [25]. The sensitivity analyses after removing the one study with NOS <7 [26] did not alter the final results in any of the genetic comparisons in the whole studied population or in Caucasians, further confirming the reliability of the results. No significant publication bias was detected in all three genetic models.

**SNP rs10757278.** The role of rs10757278 in IS was analyzed in 10 studies [20, 23, 24, 27, 28, 30, 35, 36, 40, 42] involving 9,352 cases and 2, 4552 controls. A positive association was found in the whole studied population, and in Asians and Caucasians with IS using the combined results. The G allele and GG genotype increased the susceptibility to IS in the whole studied population (G allele: OR = 1.11, 95%CI: 1.04–1.20, p-FDR = 0.006; GG genotype: OR = 1.19, 95%CI: 1.06–1.34, p-FDR = 0.006) (Figs 3 and 4), in Asians (G allele: OR = 1.16, 95%CI: 1.04–1.30, p-FDR = 0.0135; GG genotype: OR = 1.25, 95%CI: 1.07–1.48, p-FDR = 0.0135), and in Caucasians (G allele: OD = 1.12, 95%CI: 1.04–1.20, p-FDR = 0.006; GG genotype: OR = 1.18, 95%CI: 1.05–1.33, p-FDR = 0.007. The AA genotype played a protective role against IS in the whole studied population (OR = 0. 94, 95%CI: 0.88–1.00, p-FDR = 0.04), in Asians (OR = 0.91, 95%CI: 0.82–1.00, p-FDR = 0.04), and in Caucasians (OR = 0.88, 95%CI: 0.78–0.98 p-FDR = 0.021).

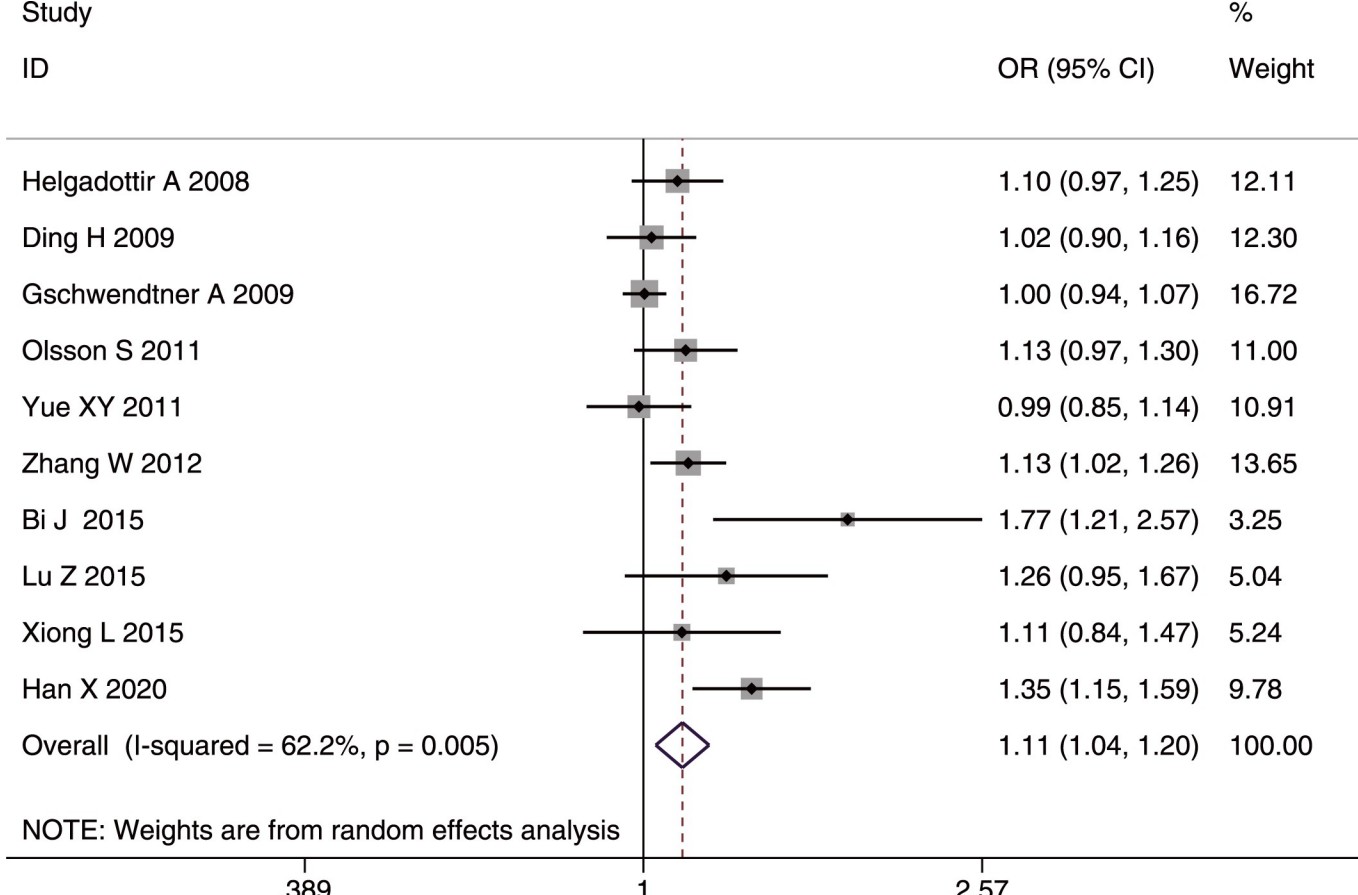

**Fig 3. Forest plot of rs10757278 allele frequency (G vs. A) in the whole studied population.**

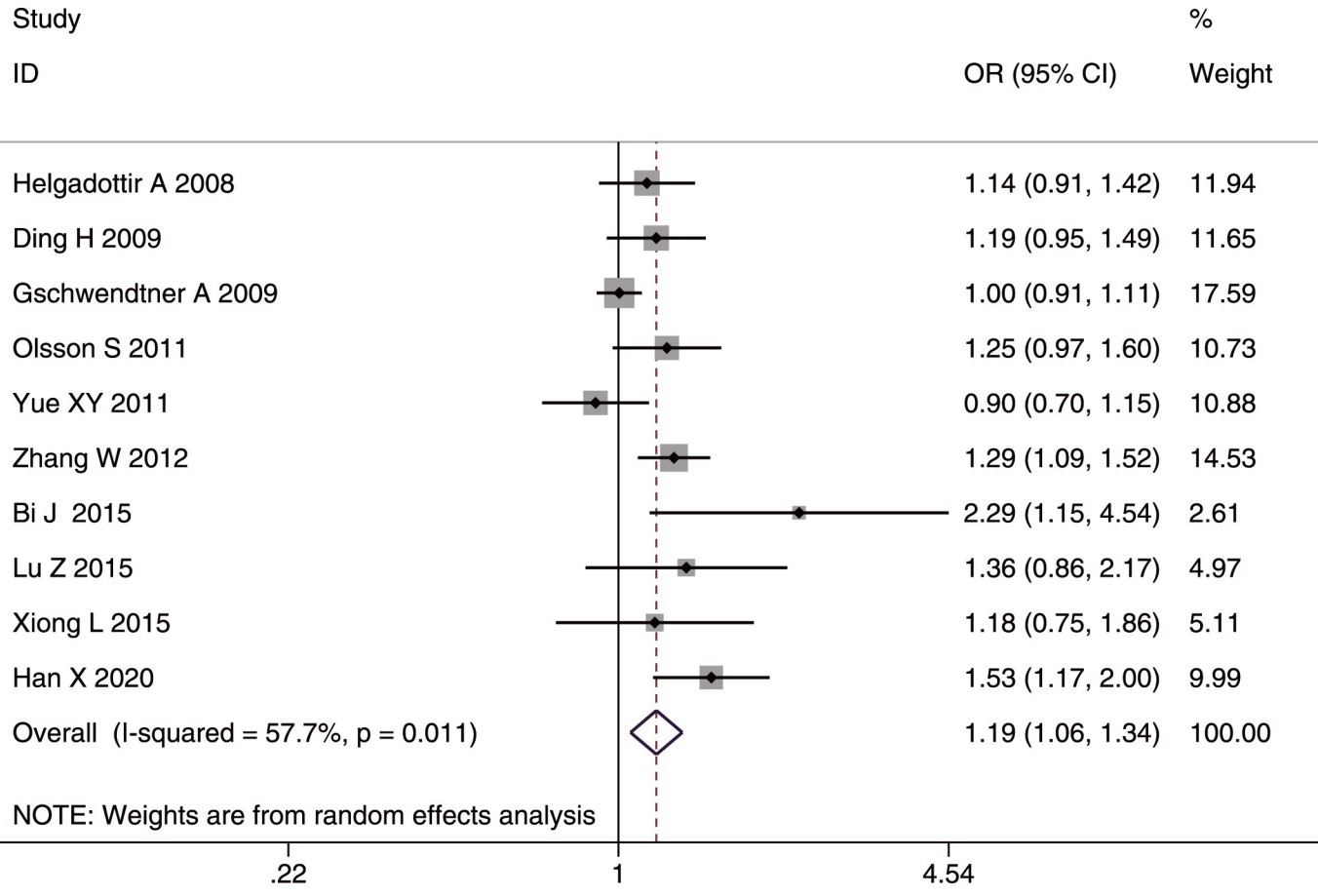

**Fig 4. Forest plot of rs10757278 genotype frequency (GG vs. (AA+GA)) in the whole studied population.**

Significant heterogeneity was found in both the allelic comparison and recessive model (GG vs. (AA+GA)) in the whole studied population (AC: $I^2$ = 62.2%, p = 0.005; RM: $I^2$ = 57.7%, p = 0.011), but mainly in Asians (AC: $I^2$ = 63.2%, p = 0.012; RM: $I^2$ = 52.3%, p = 0.05), which suggested that Asians may be the source of heterogeneity.

For IS subtypes, the G allele or GG genotype increased the risk for LAA alone in the whole studied population (G allele: OR = 1.16, 95%CI: 1.01–1.33, p-FDR = 0.038; GG genotype: OR = 1.29, 95%CI: 1.15–1.45), p-FDR = 0.000), in Asians (G allele: OR = 1.28, 95%CI: 1.05–1.56, p-FDR-0.0255; GG genotype: OR = 1.44, 95%CI: 1.22–1.71), p-FDR = 0.000), and in Caucasians (G allele: OR = 1.12, 95%CI: 1.02–1.24, p-FDR = 0.04; GG genotype: OR = 1.19, 95% CI: 1.02–1.39, p-FDR = 0.04). In contrast, the AA genotype had a protective effect on LAA in the whole studied population (OR = 0.87, 95%CI: 0.78–0.98, p-FDR = 0.0375) and in Asians (OR = 0.83, 95%CI: 0.70–0.99, p-FDR = 0.042). No heterogeneity was detected in any of the comparisons for IS subtypes. Additionally, no age difference was found in the three genetic models. The sensitivity analyses excluding the low-quality studies (NOS <7) [20, 35] did not affect the stability of the original results. We found a publication bias in the allelic comparison in the whole studied population (p = 0.019, Egger's test) (Fig 5), indicating that more studies are needed to verify the conclusion.

**SNP rs2383206.** The role of rs2383206 in IS was investigated in nine studies involving 4,431 cases and 8,423 controls) [19, 22–25, 28, 30, 35, 40]. The G allele and GG genotype

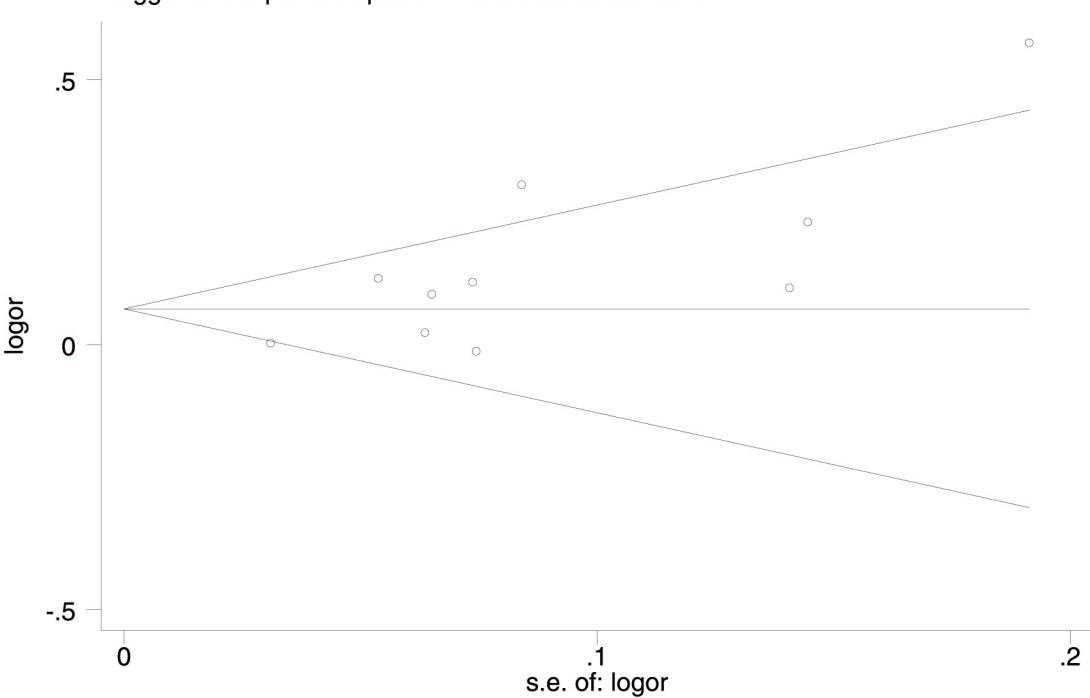

**Fig 5. Funnel plot of rs10757278 studies in the allelic comparison in the whole studied population.**

increased the IS risk in the whole studied population (G allele: OR = 1.08, 95%CI: 1.02–1.14, p-FDR = 0.0075; GG genotype: OR = 1.15, 95%CI: 1.05–1.26, p-FDR = 0.0075) (Figs 6 and 7) and in Asians (G allele: OR = 1.09, 95%CI: 1.03–1.16, p-FDR = 0.015; GG genotype: OR = 1.15, 95%CI: 1.03–1.28, p-FDR = 0.015). Three studies analyzed rs2383206 in IS sub-types, and the pooled results showed that carriers with G and GG had increased risk for the LAA subtype (G allele: OR = 1.17, 95%CI: 1.06–1.29, p-FDR = 0.0015; GG genotype: OR = 1.30, 95%CI: 1.11–1.51, p = FDR = 0.0015). In contrast, the AA genotype decreased sus-ceptibility to LAA (OR = 0.85, 95%CI: 0.73–0.99, p-FDR = 0.039). No significant association with IS was detected in the age subgroup (<45 vs. ≥45 years old). There was no heterogeneity in any of the comparisons.

The sensitivity analyses after excluding the poor-quality study [35]) gave similar overall results, confirming the stability of the results. There was no publication bias under the three genetic models in the whole studied population (Egger's test for AC p = 0.978, for DM p = 0.572, for RM p = 0.569).

**SNP rs1333040.** The role of rs1333040 in IS was analyzed in nine studies [21, 23, 24, 27, 29, 31, 32, 37, 40] involving 6,581 cases and 8,379 controls.

The combined results showed that the TT genotype conferred increased risk (OR = 1.09, 95%CI: 1.00–1.19, p-FDR = 0.044) (Fig 8), and the C allele or CC genotype played a protective role in IS in the whole studied population (C allele: OR = 0.92, 95%CI: 0.88–0.97, p-FDR = 0.003; CC genotype: OR = 0.83, 95%CI: 0.73–0.94, p-FDR = 0.006). In contrast, in the sub-population analyses, the C allele showed a protective effect on IS, but only in in Caucasians (OR = 0.92, 95%CI: 0.86–0.98, p-FDR = 0.018).

No significant relationship of rs1333040 with LAA was found in the whole studied popula-tion; however, an association with LAA risk was found in Caucasians. Patients with the C allele and CC genotype had a lower possibility of developing LAA (C allele: OR = 0.86, 95%CI: 0.76–

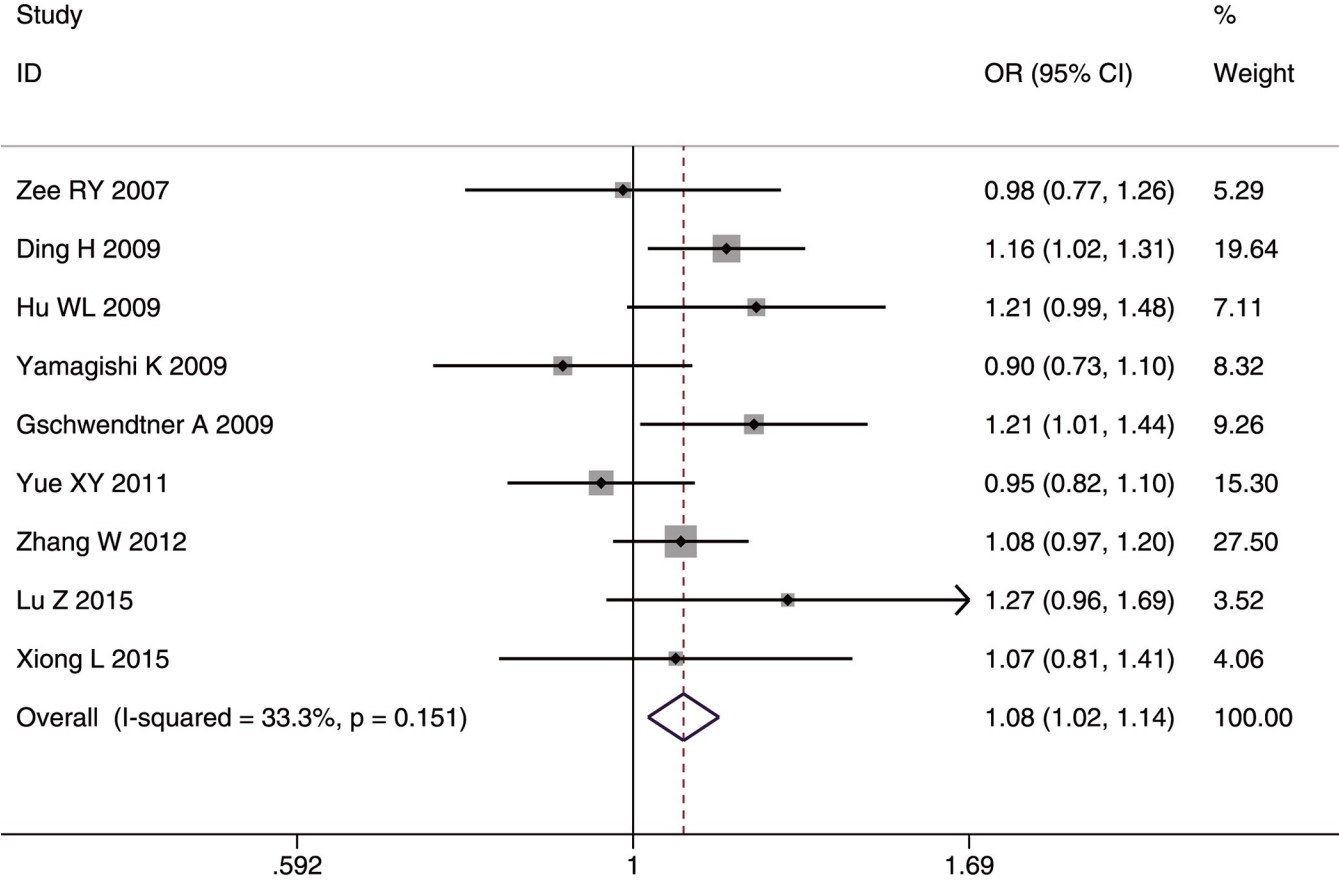

Study ID | OR (95% CI) | % Weight

| Study ID | OR (95% CI) | % Weight |
|---|---|---|
| Zee RY 2007 | 0.98 (0.77, 1.26) | 5.29 |
| Ding H 2009 | 1.16 (1.02, 1.31) | 19.64 |
| Hu WL 2009 | 1.21 (0.99, 1.48) | 7.11 |
| Yamagishi K 2009 | 0.90 (0.73, 1.10) | 8.32 |
| Gschwendtner A 2009 | 1.21 (1.01, 1.44) | 9.26 |
| Yue XY 2011 | 0.95 (0.82, 1.10) | 15.30 |
| Zhang W 2012 | 1.08 (0.97, 1.20) | 27.50 |
| Lu Z 2015 | 1.27 (0.96, 1.69) | 3.52 |
| Xiong L 2015 | 1.07 (0.81, 1.41) | 4.06 |
| Overall (I-squared = 33.3%, p = 0.151) | 1.08 (1.02, 1.14) | 100.00 |

**Fig 6. Forest plot of rs2383206 allele frequency (G vs. A) in the whole studied population.**

0.96, p-FDR = 0.03; CC genotype: OR = 0.78, 95%CI: 0.63–0.98, p-FDR = 0.037). In contrast, patents with the TT genotype seemed to be more predisposed to LAA risk (OR = 1.20, 95%CI: 1.01,1.42, P-FDR = 0.037). No sex difference was found for IS in any of the comparisons. There was no significant heterogeneity among the studies.

The sensitivity analyses after excluding low-quality studies (NOS <7) [29, 32] did not alter the final results. No publication bias was detected in the three genetic models in the whole studied population (Egger's test for AC p = 0.772, for DM p = 0.502, for RM p = 0.875).

**SNP rs1333049.** The role of rs1333049 in IS was analyzed in seven studies involving 5,351 cases and 6,061 controls [21, 23, 28, 29, 35, 39, 40]. Pooled analyses showed that the C allele increased the susceptibility to IS (OR = 1.09, 95%CI: 1.03–1.15, p-FDR = 0.009) in the whole studied population (Fig 9) and in Caucasians (OR = 1.15, 95%CI: 1.06–1.24, p-FDR = 0.001). No significant association was found in Asians, LAA subtype, or age subgroup (<45 vs. ≥45 years old). No heterogeneity was detected in any of the genetic comparisons.

The sensitivity analyses after removing low-quality studies (NOS <7) [29, 35, 39] remained unchanged in the three models in the whole studied population. No publication bias was found in three genetic models (Egger's test for AC p = 0.845, for DM p = 0.854, for RM p = 0.187).

**SNP rs1537378.** The role of rs1537378 in IS was analyzed in six studies [23, 27, 28, 35, 36, 40] involving 6,166 cases and 6,129 controls. The CC genotype was found to increase the risk for IS in the whole studied population (OR = 1.18, 95%CI: 1.09–1.27, p-FDR = 0.000) (Fig 10),

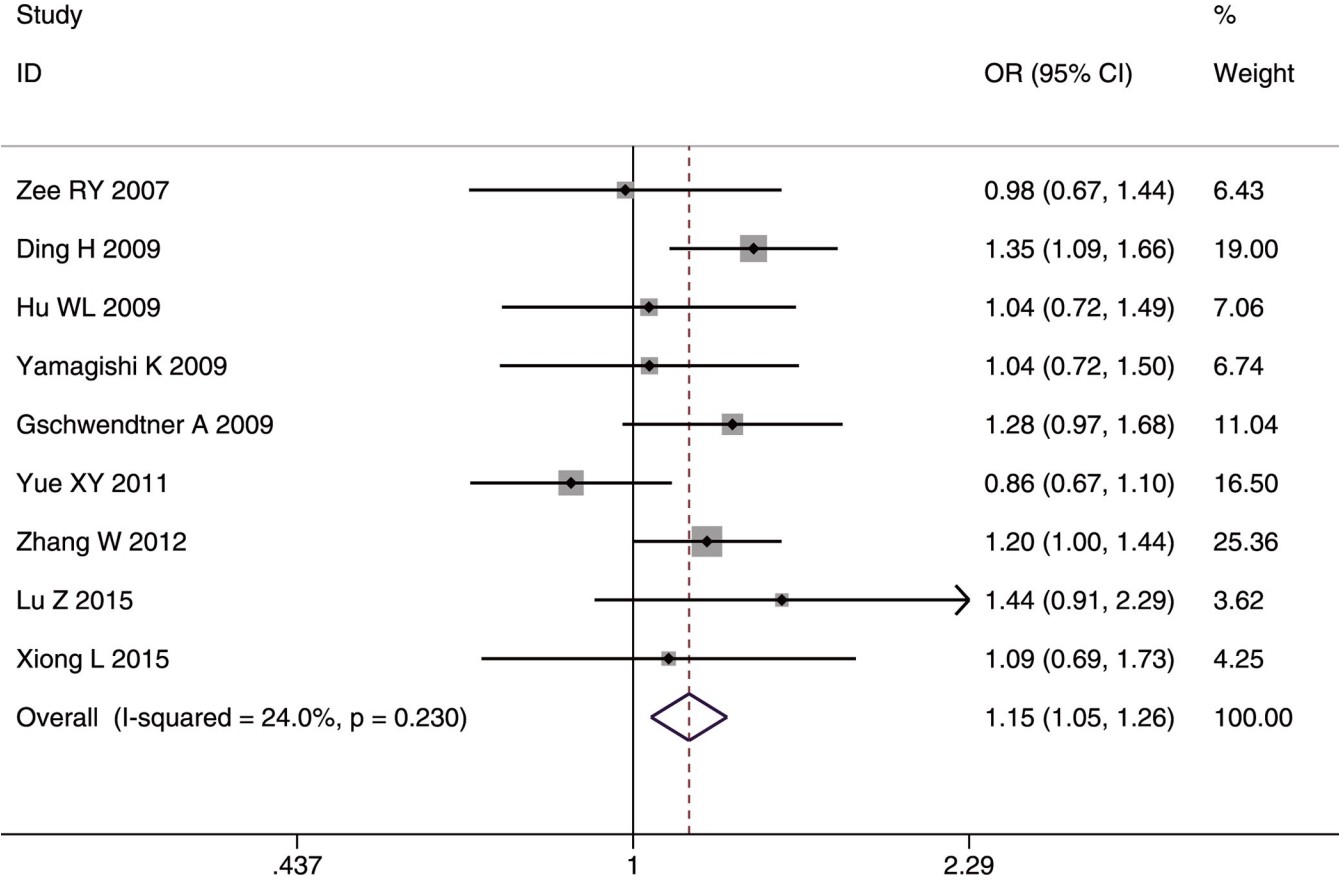

**Fig 7. Forest plot of rs2383206 genotype frequency (GG vs. (AA+AG)) in the whole studied population.**

in Asians (OR = 1.43, 95%CI: .1.20–1.71, p-FDR = 0.000), and in Caucasians (OR = 1.21, 95% CI: 1.07–1.37, p-FDR = 0.003). In contrast, the T allele and TT genotype had a protective effect on IS in the whole studied population (T allele: OR = 0.80, 95%CI: 0.70–0.92, p-FDR = 0.001; TT genotype: OR = 0.83, 95%CI: 0.74–0.93, p-FDR = 0.001), in Asians (T allele: OR = 0.70, 95%CI: 0.60–0.82, p-FDR = 0.000; TT genotype: OR = 0.49, 95%CI: 0.30–0.80, p-FDR = 0.005), and in Caucasians (T allele: OR = 0.85, 95%CI: 0.78–0.93, p-FDR = 0.000; TT genotype: OR = 0.79, 95%CI; 0.67–0.93, p-FDR = 0.006).

In the IS subtype analyses, a significant relationship was found only in LAA. The LAA risk was higher in carriers with the CC genotype, and patients carrying the T allele and TT genotype had lower risk for LAA in the whole studied population, in Asians, and in Caucasians. In patients who were ≥45 years old, the CC genotype was also associated with higher risk for all types of IS, and only T allele had a protective role.

Significant heterogeneity among studies was found in the T allele (T/C) and CC genotype comparisons (CC vs. (CT+TT)) only in the whole studied population. The heterogeneity disappeared after removing the study by Bi *et al.* [36], which suggested it may be a source of heterogeneity; however, the final results remained unchanged. The sensitivity analyses after excluding the study with NOS = 6 [35] did not alter any of the results, indicating the reliability and stability of the original results. The funnel plot was asymmetric in all three genetic comparisons in the whole studied population (Egger's test for AC p = 0.019; for DM p = 0.033; for RM p = 0.046) (Fig 11), which suggested there might be some publication bias. The trim and

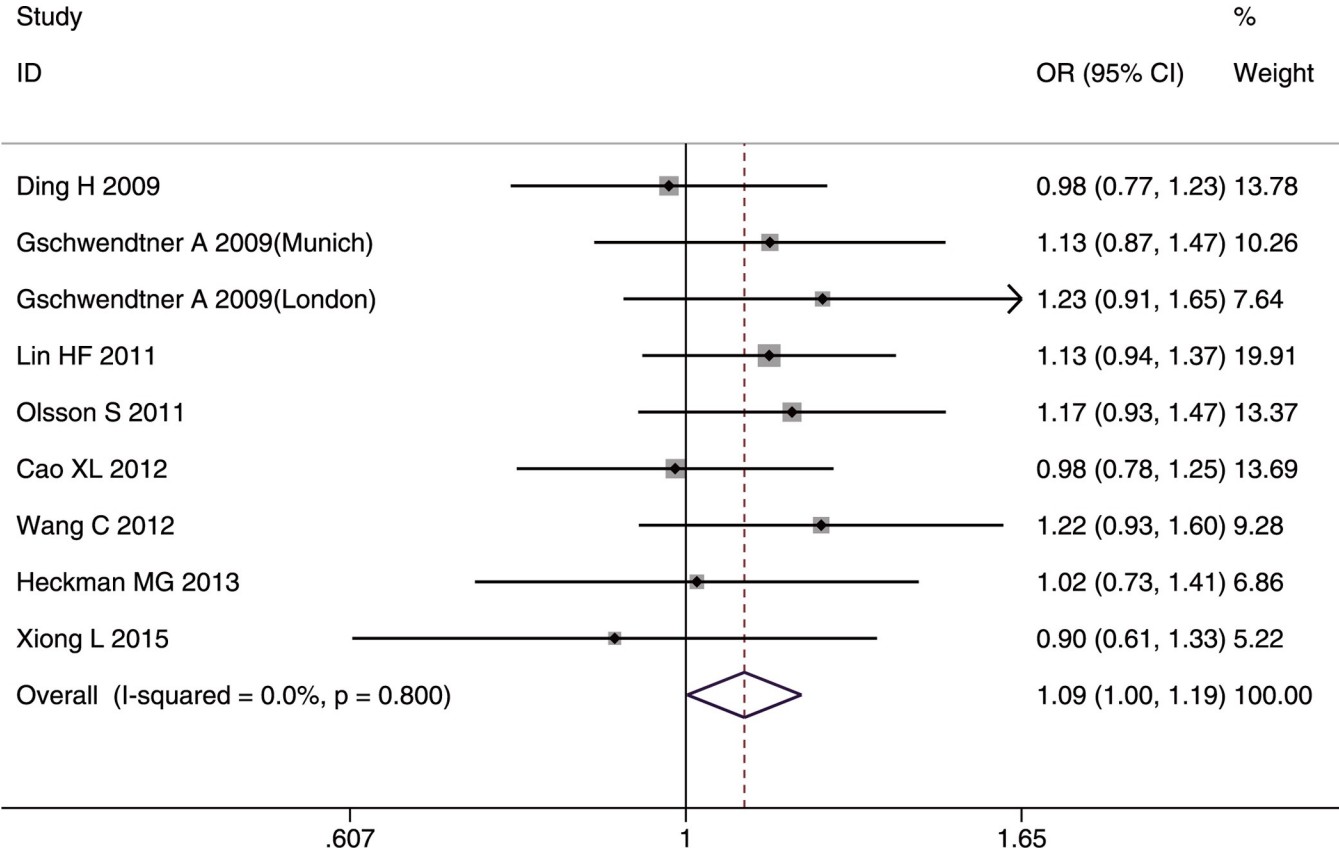

**Fig 8. Forest plot of rs1333040 genotype frequency (TT vs. (CC+CT)) in the whole studied population.**

fill method was used to identify and correct the bias, and the combined effect was found to be unchanged, indicating that the possible publication bias had little effect on the results.

**SNP rs4977574.** The role of rs4977574 in IS was analyzed in five studies [32, 33, 35, 37, 43] involving 6,083 cases and 4,593 controls that included three Asian, one Caucasian, and one mixed populations.

The pooled results indicated that rs4977574 was strongly associated with IS. It was found that The G allele and GG genotype conferred susceptibility IS in the whole studied population (G allele: OR = 1.11, 95%CI: 1.05–1.17, p-FDR = 0.000; GG genotype: OR = 1.13, 95%CI: 1.03–1.24, p-FDR = 0.011) (Figs 12 and 13). In contrast, the AA genotype decreased the risk of IS in the whole studied population (OR = 0.86, 95%CI: 0.79–0.94, p-FDR = 0.0015).

The G allele and AA genotype had significant association with IS risk only in Asians (G allele: OR = 1.20, 95%CI: 1.06–1.36, p-FDR = 0.0004; AA genotype: OR = 0.75, 95%CI: 0.63–0.89, p-FDR = 0.0015). Significant heterogeneity was found only in the allelic comparison model ($I^2$ = 64.1%, p = 0.062) in Caucasians; however, the heterogeneity disappeared ($I^2$ = 0%, p = 0.714) after removing the study by Lovkvist *et al.* [33].

The IS subtype analysis showed that the G allele and GG genotype were risk factors for LAA in the whole studied population (G allele: OR = 1.22, 95%CI:1.09–1.37, p-FDR = 0.003; GG genotype: OR = 1.26, 95%CI:1.05–1.52, p-FDR-0.015) and in Caucasians (G allele: OR = 1.26, 95%CI:1.09–1.46, p-FDR = 0.006; GG genotype: OR = 1.35, 95%CI:1.07–1.71, p-FDR = 0.014). In contrast, the AA genotype had a protective role in the whole studied

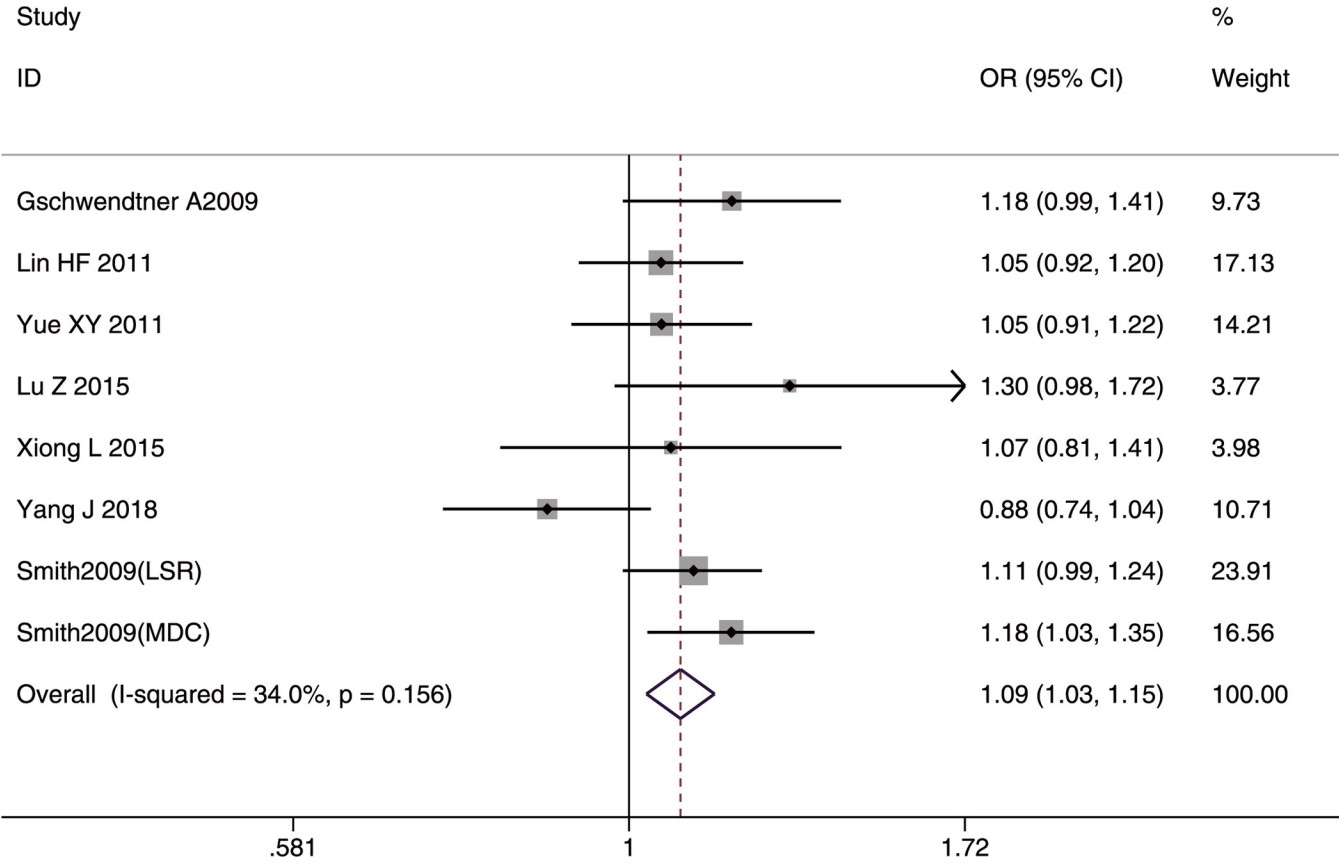

**Fig 9. Forest plot of rs1333049 allele frequency (C vs. G) in the whole studied population.**

population (OR = 0.75, 95%CI:0.62–0.90, p-FDR = 0.003) and in Caucasians (OR = 0.74, 95% CI:0.58–0.94, p-FDR = 0.014). Heterogeneity was detected in the small-vessel occlusion and cardioembolism subtypes; however, the source of the heterogeneity was not analyzed because of the small number of included studies.

The sensitivity analysis after omitting two poor-quality studies [32, 35] showed that the final pooled results were not affected. No publication bias was detected by the funnel plots or Egger's test in the three genetic models in the whole studied population.

**SNP rs1004638.** The role of rs1004638 in IS was analyzed in three studies [24, 28, 40] comprising only subjects with 1,959 cases and 1,941 controls. Significant associations of this SNP with IS were found in all genetic comparisons (AC: OR = 1.15, 95%CI: 1.04–1.26, p-FDR = 0.015; RM: OR = 1.21, 95%CI: 1.03–1.43, p-FDR = 0.024; DM: OR = 0.85, 95%CI: 0.73–0.98, p-FDR = 0.024) without no heterogeneity among the studies. The A allele and AA genotype increased susceptibility to IS, whereas the TT genotype had a protective role. Sensitivity analysis and publication bias were not performed because of the small number of included studies.

**Other SNPs.** For each of the remaining six SNPs, rs7865618, rs10965227, rs1333042, rs7044859, rs10116277, and rs10757269, only two studies with from 512 to 4,322 cases and from 752 to 4,477 controls, were included for meta-analyses. No significant association was found in any of the comparisons. Heterogeneity between studies, sensitivity analysis, and publication bias were not explored because of the small number of studies for each SNP.

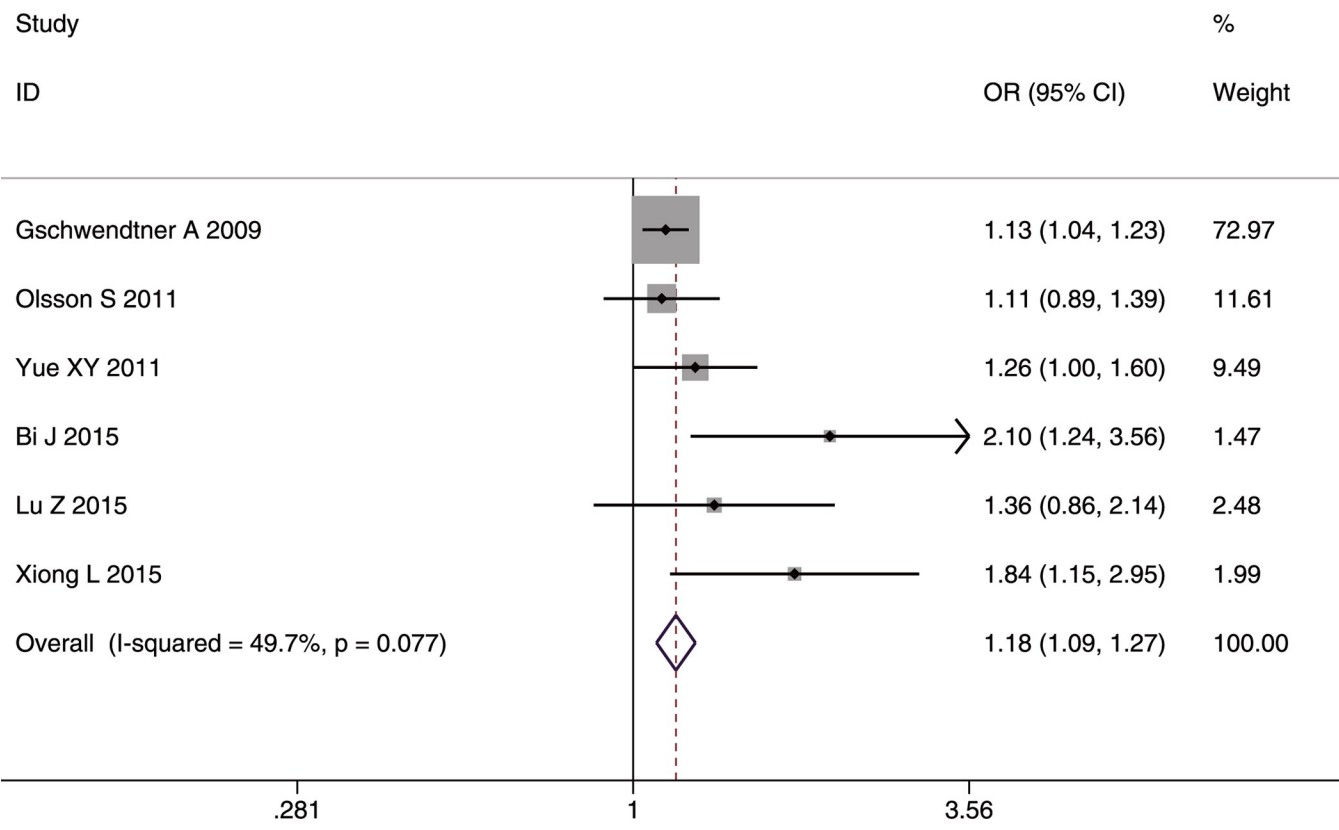

**Fig 10. Forest plot of rs1537378 genotype frequency (CC vs. (CT+TT)) and susceptibility to all types of IS in the whole studied population.**

## Discussion

The meta-analysis results showed that eight SNPs (rs10757274, rs10757278, rs2383206, rs1333040, rs1333049, rs1537378, rs4977574, and rs1004638) in *ANRIL* were significantly associated with IS risk, and six of these SNPs (rs10757274, rs10757278, rs2383206, rs1333040, rs1537378, and rs4977574) were also found to be related to the LAA subtype of IS. Two of the SNPs (rs2383206 and rs4977574) were associated with IS mainly in Asians, and three SNPs (rs10757274, rs1333040, and rs1333049) were associated with susceptibility to IS mainly in Caucasians.

The locus close to a cluster of cell-cycle regulating genes in chromosome 9p21, such as *CDKN2A* and *CDKN2B*, regulates vascular remodeling pathways. The proteins encoded by these genes affect cell-cycle progression, resulting in an antiproliferative effect on arterial smooth muscle. In human white blood cells, the homozygous carriers of the 9p21 risk allele are associated with down-regulation of *CDKN2B* expression and up-regulation of genes involved in cellular proliferation. Markedly decreased expression of *CDKN2A* and *CDKN2B* was reported in mutant mice and doubling of the proliferative capacity of mutant aortic smooth muscle cells in culture was detected, a cellular phenotype relevant to atherosclerosis [44].

*ANRIL* encodes a large antisense long non-coding RNA in which the first exon is located in the *CDKN2A* promoter and overlaps with the two exons of *CDKN2B*. Expression of *ANRIL* co-clustered mainly with p14/ARF under both physiologic and pathologic conditions. The 9p21 region may promote atherosclerosis by regulating the expression of *ANRIL*, which in turn is associated with altered expression of genes that control cellular proliferation pathways [9].

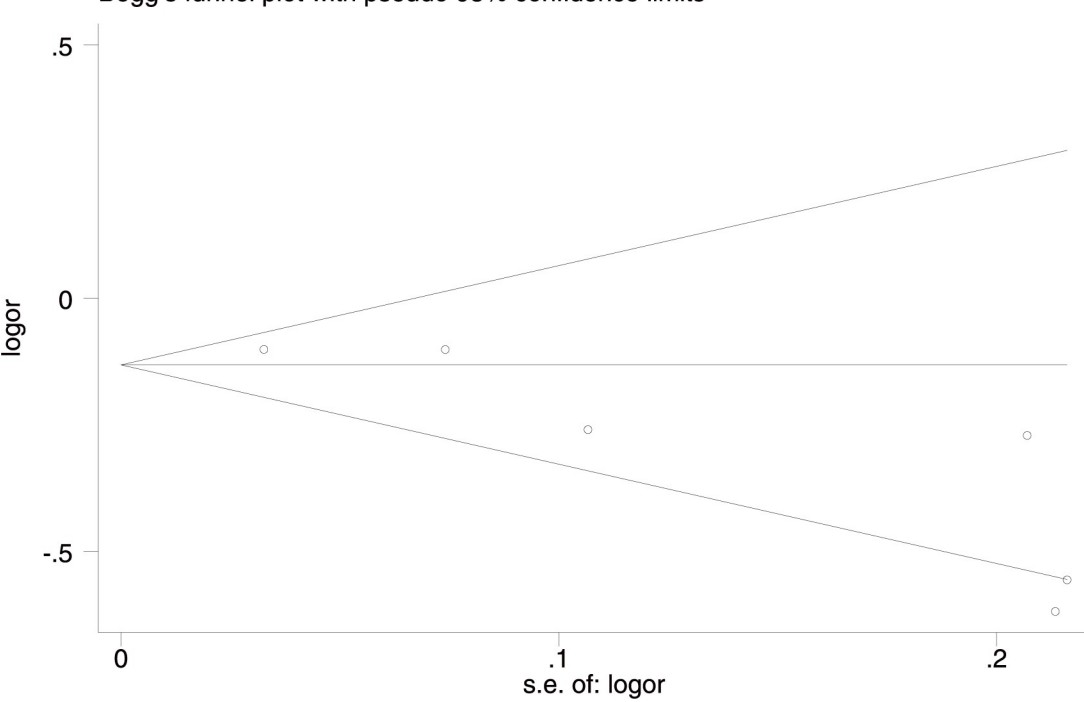

Begg's funnel plot with pseudo 95% confidence limits

**Fig 11. Funnel plot of rs1537378 allele frequency (T vs. C) and IS susceptibility in the whole studied population.**

*ANRIL* was recently shown to be expressed in human atheromatous vessels, including both abdominal aortic aneurysm and carotid endarterectomy samples, as well as in isolated vascular endothelial cells, monocyte-derived macrophages, and coronary smooth muscle cells. Moreover, *ANRIL* expression was significantly associated with the alteration in function of vascular endothelial cells and vascular smooth muscle cells in both human or animal models [45]. Together, these findings indicate that *ANRIL* has a direct effect on the pathobiology of atherosclerosis. Therefore, *ANRIL* is considered a good candidate for atherosclerotic disease risk, such as coronary artery disease (CAD) and IS [46, 47].

Studies have shown that different *ANRIL* transcripts exhibit disease-specific expression patterns in CAD, which further supports the hypothesis that *ANRIL* is the causative gene at the 9p21 CAD susceptibility locus [48]. Recently, a few meta-analyses using SNPs also indicated a significant association of *ANRIL* with CAD [49–55]. IS is known to share common pathophysiological mechanisms with CAD, and CAD and IS seem to have common susceptibility locus. A comprehensive review indicated that increased *ANRIL* expression was associated with IS risk in animal models by promoting angiogenesis and regulating inflammation [56], and patients with IS were also found to have significantly higher serum *ANRIL* levels in clinical practice [57, 58].

Some studies have explored the functional effect of SNPs in *ANRIL*. The rs1333049 risk allele (C allele) was found to influence *ANRIL* expression levels in vascular smooth muscle cells, which was associated with elevated levels of these cells in atherosclerosis plaques involved in the pathogenesis of atherosclerosis [59]. Rs1333040 is located in an intronic enhancer region that was found to influence the activity of the enhancer and *ANRIL* expression. Rs10757274 showed high linkage disequilibrium with myocardial infraction-associated SNPs, including rs1537373, rs4977575, and rs10757272, and contributed to the activation or inhibition of the expression of the related genes [55]. A few SNPs were found to have a significant relationship

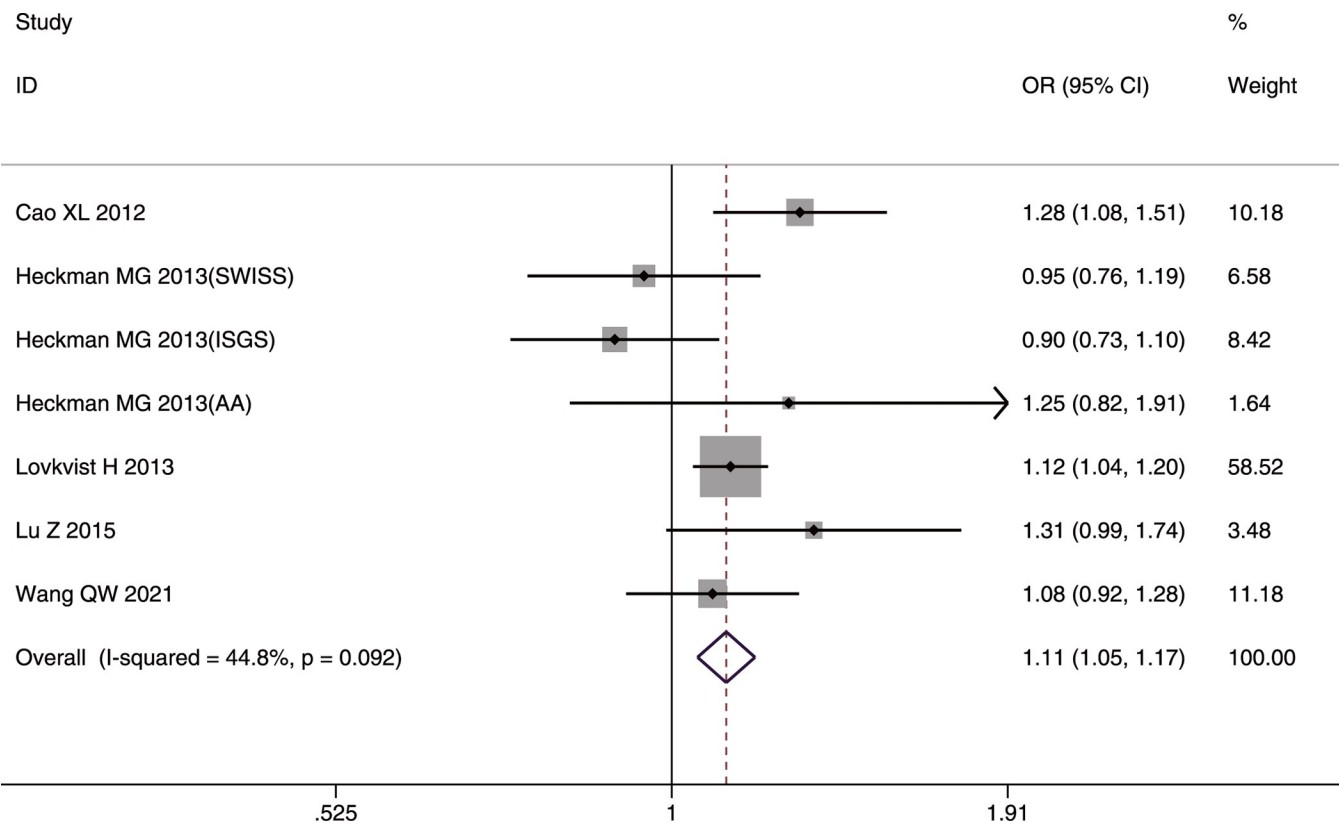

**Fig 12. Forest plot of rs4977574 allelic frequency (G vs. A) and IS susceptibility in the whole studied population.**

to vascular risk factors. Patients carrying mutant alleles of rs1333049 and rs4977574 had elevated total cholesterol, triglycerides, and low-density lipoprotein cholesterol levels [60–62]. The risk allele of rs4977574 was also found to be related to carotid plaque formation in patients with acute IS [63] or type 2 diabetes [64]. All of these factors may lead to the progression of atherosclerotic vascular diseases or IS.

A few meta-analyses have reported the association of *ANRIL* with IS; however, these meta-analyses have some limitations, such as failure to include all eligible studies [34, 43, 52, 65–68], no comprehensive analyses [66–68], confounding cases (patients with transient ischemic attack or other types of stroke were included in the IS samples) [34, 52, 65, 67], as well as wrong SNP loci [65] or errors in extracting and analyzing data [34, 65, 67], which could have influenced the overall results. Two previous genome-wide association studies (GWAS) [69, 70] explored the relationship of *ANRIL* SNPs and IS in a Caucasian cohort with European ancestry, but only one SNP (rs2383207) was found to be association with LAA. Ethnicity may partly explain the discrepancy between the GWAS results and the results of the present meta-analysis, which included more Asians.

The potential biological mechanisms, including how *ANRIL* is strongly associated with the risk for cardio-metabolic diseases, are still unknown. Recent reports have found that the N4-acetylcytidine modification of RNA, which regulated gene expression, and microRNA-mediated gene expression and immuno-deficiency in the gut microbiome, were key to cardio-metabolic diseases, including IS [71–78]. However, the few studies that have investigated the role of *ANRIL* SNP loci in the N4-acetylcytidine regulatory pathway failed to find definite effects of RNA modification or immuno-deficiency on the development of IS.

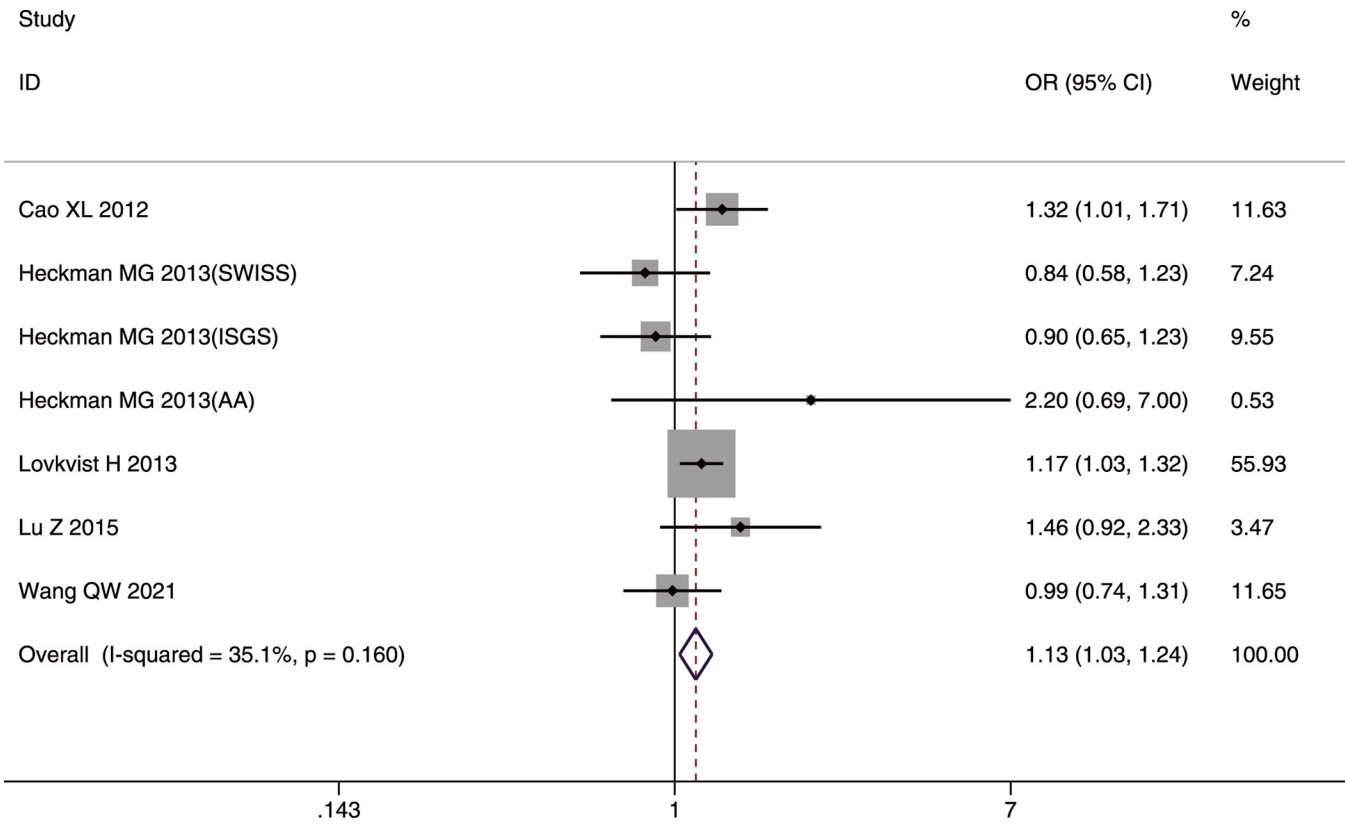

**Fig 13. Forest plot of rs4977574 genotype frequency (GG vs. (GA+AA)) and IS susceptibility in the whole studied population.**

Our meta-analysis has some limitations. Firstly, there is language bias because we only searched studies of *ANRIL* polymorphisms on IS reported in Chinese and English, and therefore may have missed studies published in other languages. Secondly, the number of studies included in this meta-analysis was moderate, and seven of the SNPs (rs1004638, rs7865618, rs10965227, rs1333042, rs7044859, rs10116277, and rs10757269) were involved in three or less studies. Therefore, some results could be influenced by random error and/or publication bias. Thirdly, the presence of potential confounders between studies or between cases and controls within each study, such as age, sex, or ethnic admixture, were unadjusted that may have influenced the results. Fourthly, it is well known that it is very important to conduct causal inference analysis to determine if the associated genetic polymorphisms are causally triggering the development of IS by mediating the expression of this gene in specific tissues [79–82]. Although, this meta-analysis aimed to discuss the association of *ANRIL* with IS using SNPs as genetic marker, no causal genetic effects of *ANRIL* on IS can be established. Fifthly, machine learning is considered a useful tool for the classification and prediction of diseases based on biomarkers [83–86] that we have yet to use to analyze the role of *ANRIL* in susceptibility to IS. Sixthly, GWAS, case-only studies, and family-based studies were not included because of differences in study design, but they could be useful for meta-analysis in the future. Finally, the inter-study heterogeneity in the pooled analyses may have affected the results for several SNPs.

In summary, our accumulated pooled analyses indicate that *ANRIL* has a significant association with IS risk in Asian populations. The causal effects of the *ANRIL* SNPs associated with IS can be explored by Mendelian randomization analysis in the future.

## Supporting information

**S1 Appendix. PRISMA 2009 checklist used in this meta-analysis.**
(DOCX)

**S2 Appendix. Meta-analysis on genetic association studies checklist.**
(DOCX)

**S3 Appendix. The excluded articles and the reasons for exclusion of each article.**
(DOCX)

## Acknowledgments

We thank Dr. Qiwen Mu from North Sichuan Medical College for helpful comments on the manuscript. We thank Margaret Biswas, PhD, from Liwen Bianji (Edanz) (www.liwenbianji. cn/) for editing the English text of a draft of this manuscript.

## Author Contributions

**Conceptualization:** Hua Liu.

**Data curation:** Na Bai, Wei Liu.

**Formal analysis:** Na Bai, Wei Liu, Tao Xiang, Qiang Zhou, Jun Pu, Jing Zhao, Danyang Luo, Xindong Liu.

**Funding acquisition:** Hua Liu.

**Investigation:** Na Bai, Wei Liu, Tao Xiang.

**Methodology:** Na Bai, Wei Liu, Tao Xiang, Qiang Zhou, Jun Pu, Jing Zhao, Danyang Luo, Xindong Liu, Hua Liu.

**Project administration:** Na Bai, Wei Liu, Hua Liu.

**Resources:** Tao Xiang.

**Software:** Na Bai, Wei Liu.

**Supervision:** Hua Liu.

**Validation:** Na Bai, Wei Liu, Hua Liu.

**Visualization:** Na Bai, Wei Liu, Hua Liu.

**Writing – original draft:** Na Bai, Wei Liu, Hua Liu.

**Writing – review & editing:** Hua Liu.

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
