## [Decision Letter · Decision Letter 0]

13 Sep 2021

PONE-D-21-27176The Research on Association of the ANRIL Gene with Ischemic Stroke: the Evidence From a Comprehensive Meta-AnalysisPLOS ONE

Dear Dr. Liu,

Thank you for submitting your manuscript to PLOS ONE. After careful consideration, we feel that it has merit but does not fully meet PLOS ONE’s publication criteria as it currently stands. Therefore, we invite you to submit a revised version of the manuscript that addresses the points raised during the review process. The title should have a clear, precise scientific meaning and should not contain a colon. Where possible, the title should be read as one concise sentence. Please re-write the title ensuring that it is informative and appropriate. Please submit your revised manuscript by September 20. If you will need more time than this to complete your revisions, please reply to this message or contact the journal office at plosone@plos.org. Please include the following items when submitting your revised manuscript:A rebuttal letter that responds to each point raised by the academic editor and reviewer(s). You should upload this letter as a separate file labeled 'Response to Reviewers'.A marked-up copy of your manuscript that highlights changes made to the original version. You should upload this as a separate file labeled 'Revised Manuscript with Track Changes'.An unmarked version of your revised paper without tracked changes. You should upload this as a separate file labeled 'Manuscript'.

We look forward to receiving your revised manuscript.

Kind regards,

Mingqing Xu

Academic Editor

PLOS ONE

Journal Requirements:

Additional Editor Comments:

The title should have a clear, precise scientific meaning and should not contain a colon. Where possible, the title should be read as one concise sentence. Please re-write the title ensuring that it is informative and appropriate.

Reviewers' comments:

Reviewer's Responses to Questions

**Comments to the Author**

1. Is the manuscript technically sound, and do the data support the conclusions?

Reviewer #1: Yes

Reviewer #2: No

2. Has the statistical analysis been performed appropriately and rigorously? 

Reviewer #1: Yes

Reviewer #2: No

3. Have the authors made all data underlying the findings in their manuscript fully available?

Reviewer #1: No

Reviewer #2: Yes

4. Is the manuscript presented in an intelligible fashion and written in standard English?

Reviewer #1: No

Reviewer #2: No

5. Review Comments to the Author

Reviewer #1: In the manuscript entitled “The Research on Association of the ANRIL Gene with Ischemic Stroke: the Evidence From a Comprehensive Meta-Analysis”, the authors explored whether f the All the data used in the published GWASA papers should be included, if cannot, please state the reasons.

The following papers can be cited and followed for the meta-analytic procedures to improve the quality. (if the data is not enough available, at least DISCUSSION should be added as the LIMITATION of this study with enough citation to support the viewpoints):

Ref 1: Wu Y, et al. Multi-trait analysis for genome-wide association study of five psychiatric disorders. Transl Psychiatry. 2020 Jun 30;10(1):209.

Ref 2: Jiang L, et al. Sex-Specific Association of Circulating Ferritin Level and Risk of Type 2 Diabetes: A Dose-Response Meta-Analysis of Prospective Studies..J Clin Endocrinol Metab. 2019 Oct 1;104(10):4539-4551.

Ref 3: Xu M, Lin Z. Genetic influences of dopamine transport gene on alcohol dependence: a pooled analysis of 13 studies with 2483 cases and 1753 controls.Prog Neuropsychopharmacol Biol Psychiatry. 2011 Jul 1;35(5):1255-60.

Ref 4: Xu M, Sham P, Ye Z, Lindpaintner K, He L. A1166C genetic variation of the angiotensin II type I receptor gene and susceptibility to coronary heart disease: collaborative of 53 studies with 20,435 cases and 23,674 controls.Atherosclerosis. 2010 Nov;213(1):191-9.

Ref 5: Xu M, et al. Quantitative assessment of the effect of angiotensinogen gene polymorphisms on the risk of coronary heart disease. Circulation. 2007 Sep 18;116(12):1356-66

Trans-ethnitic meta-analysis can be referred to Ref 1.

Subgroup analyses based on sex, age, race, gene dosage can be referred to References 2. and Quality assessment score can also to referred to Refences 3-5.

I am wondering if the authors may integrate genotype data with eQTL from GTEX or pQTLs is to explore the causality of the genetic variant in the development of stroke. But I strongly suggest to do causal inference analysis to see if the associated Genetic Polymorphisms in this gene are causally triggering the development of stroke through mediating the expression of this gene in specific tissues. If cannot, please discuss the limitations in the Discussion in detail with additional citations to support the viewpoints. For these reasons, the following papers regarding causal inference between genetic varients,inter-mediator phenotype and disease outcome can be cited and followed.

Reference 1: Fuquan Zhang, Ancha Baranova, Chao Zhou, et al. Causal influences of neuroticism on mental health and cardiovascular disease. Human Genetics. 2021 May 1

Reference 2:Fuquan Zhang, et al. Genetic evidence suggests posttraumatic stress disorder as a subtype of major depressive disorder. Journal of Clinical Investigation. 2021 May 30

Reference 3:Wang, X. et al. Genetic support of a causal relationship between iron status and type 2 diabetes: a Mendelian randomization study. The Journal of clinical endocrinology and metabolism, doi:10.1210/clinem/dgab454 (2021).

I am not sure if the genetic polymorphism can be used for predict stroke based on a machine learning model. In the PRECISION MEDICINE era, deep learning or machine learning is a hot topic in classification and prediction of diseases based on biomarkers. The authors may discuss the possibility to use the genetic variants related to stroke for the prediction or early diagnosis of stroke. The authors may cite the following papers for discussion or follow the analytic procedures to construct machine learning prediction models.

Reference 1:Yu H, et al. LEPR hypomethylation is significantly associated with gastric cancer in males.Exp Mol Pathol. 2020 Oct;116:104493.

Reference 2:Liu M, et al. A multi-model deep convolutional neural network for automatic hippocampus segmentation and classification in Alzheimer's disease.Neuroimage. 2020 Mar;208:11645

I suggest one paragraph in the DISCUSSION section to elucidate the potential biological regulation mechanisms regarding how the genetic variant to affect the stroke outcome. The following papers clearly disclosed some genes indications whose abnormal expressions are mediated through mRNA modifications. The recent progress in N4-Acetylcytidine on RNA expression is also playing key role on the human diseases. I suggest the authors discussing this mRNA modifications/ N4-Acetylcytidine with their findings in the DISCUSSION section because the knowledge needs to be updated.

Reference1: Jin G, et al. The Processing, Gene Regulation, Biological Functions, and Clinical Relevance of N4-Acetylcytidine on RNA: A Systematic Review. Mol Ther Nucleic Acids. 2020. PMID: 32171170

Reviewer #2: 1. I am not sure if the 15 SNPs in ANRIL Gene are tagSNPs or not. Only tagSNPs that are unrelated with each other are informative, and it is better that the tagSNPs may cover this whole gene region.

2. Multiple test correction should be conducted .

3. Statiscial power should be calculaed with proper method to ensure less type 2 errors.

3. subgroup analysis based on sex, age should be conducted.

4. I would suggest collecting published GWAS data to redo the meta-analysis if it is possible.

5. For the genetic varients with significant association with the phenotype, deep discussion about the potential biological mechanisms involved in the phenotype is need. Especially, you had better see if this kind of SNP is causal or not.. Especially the potential causal effects for the SNPs with strong association signals should be explored.

All the figures are not clear and need to be well tailored for publishing.

The language should be polished further.

6. PLOS authors have the option to publish the peer review history of their article (what does this mean?). If published, this will include your full peer review and any attached files.

Reviewer #1: No

Reviewer #2: No

---

## [Author Response · Author response to Decision Letter 0]

14 Jan 2022

Dear editors, 

Thank you for these valuable comments, we have had a detailed answer to some problems. However, we had some difficult in dealing with other suggestions the reviewers mede. We hope that these faults will not affect the final decision the journal made. Please see this as follows:

The title should have a clear, precise scientific meaning and should not contain a colon. Where possible, the title should be read as one concise sentence. Please re-write the title ensuring that it is informative and appropriate.

Answer: the new title: Genetic association of ANRIL with susceptibility to Ischemic Stroke: a comprehensive meta-analysis

Reviewer #1:

1. I am not sure if the 15 SNPs in ANRIL Gene are tagSNPs or not. Only tagSNPs that are unrelated with each other are informative, and it is better that the tagSNPs may cover this whole gene region.

Answer: the objectives of the manuscript was all SNPs in ANRIL gene involving relationship to ischemic stroke, and aimed to investigate the role of ANRIL gene using SNPs as genetic marker in ischemic stroke risk. The SNPs were included in the meta-analysis only based on the SNP locus being studied in at least two articles. And so, it is not important whether the SNPs are tagSNPs.

2. Multiple test correction should be conducted .

Answer: We have discussed the focus in “Statistical analyses” section: “The false discovery rate (FDR) method by Benjamini–Hochberg was used for multiple testing correction”.

3. Statiscial power should be calculaed with proper method to ensure less type 2 errors.

Answer: In this meta-analysis, pooled ORs with corresponding 95% CI were calculated by the fixed-effects or the random-effects model based on three genetic models. This is a most common methods, and it is not important to calculate the statistical power again.

3. subgroup analysis based on sex, age should be conducted.

Answer: we have clearly described the proplem. Please see “Sub-population analyses were conducted for ethnicity, and meanwhile, subgroup analyses for IS subtype, age or gender (if available) were also performed” paragraph in “Statistical analyses” section.

4. I would suggest collecting published GWAS data to redo the meta-analysis if it is possible. 

Answer: we only studied the SNPs in ANRIL gene in current manuscript. The GWAS data, case-only studies and family-based studies were not included because of different study design, as well as different analysis methods might be needed in meta-analysis based on GWAS data.

5. For the genetic varients with significant association with the phenotype, deep discussion about the potential biological mechanisms involved in the phenotype is need. Especially, you had better see if this kind of SNP is causal or not.. Especially the potential causal effects for the SNPs with strong association signals should be explored.

Answer: we have tried our best to make deep discussion on the potential biological mechanisms of these risk SNPs loci, however, some biological mechanisms for a few SNPs still is not available due to too few articles involving the loci.

6, All the figures are not clear and need to be well tailored for publishing.

Answer: we have revised the figures again.

7,The language should be polished further.

Answer: we have reviewed the language problems with the help of native English speaker.

Reviewer #2:

In the manuscript entitled “The Research on Association of the ANRIL Gene with Ischemic Stroke: the Evidence From a Comprehensive Meta-Analysis”, the authors explored whether f the All the data used in the published GWASA papers should be included, if cannot, please state the reasons.

The following papers can be cited and followed for the meta-analytic procedures to improve the quality. (if the data is not enough available, at least DISCUSSION should be added as the LIMITATION of this study with enough citation to support the viewpoints):

I am not sure if the genetic polymorphism can be used for predict stroke based on a machine learning model. In the PRECISION MEDICINE era, deep learning or machine learning is a hot topic in classification and prediction of diseases based on biomarkers. The authors may discuss the possibility to use the genetic variants related to stroke for the prediction or early diagnosis of stroke. The authors may cite the following papers for discussion or follow the analytic procedures to construct machine learning prediction models.

I suggest one paragraph in the DISCUSSION section to elucidate the potential biological regulation mechanisms regarding how the genetic variant to affect the stroke outcome. The following papers clearly disclosed some genes indications whose abnormal expressions are mediated through mRNA modifications. The recent progress in N4-Acetylcytidine on RNA expression is also playing key role on the human diseases. I suggest the authors discussing this mRNA modifications/ N4-Acetylcytidine with their findings in the DISCUSSION section because the knowledge needs to be updated.

Answer: the manuscript aimed to discuss the relationship of the ANRIL gene using SNPs as genetic marker in Ischemic stroke employing the meta analyses methods. The conclusions might be a association of the targeted gene with Ischemic stroke being detected or not, however, the results did not make us find some causal genes.

The GWAS data, case-only studies and family-based studies were not included because of different study design, as well as different analysis methods might be needed in meta-analysis based on GWAS data.

 we have tried our best to make deep discussion on the potential biological mechanisms of these risk SNPs loci, however, some biological mechanisms for a few SNPs still is not available due to too few articles involving the loci.

Regards,

Hua Liu

---

## [Editor Report · Decision Letter 1]

20 Jan 2022

Genetic association of ANRIL with susceptibility to Ischemic Stroke: a comprehensive meta-analysis

PONE-D-21-27176R1

Dear Dr. Liu,

We’re pleased to inform you that your manuscript has been judged scientifically suitable for publication and will be formally accepted for publication once it meets all outstanding technical requirements.

Kind regards,

Mingqing Xu

Academic Editor

PLOS ONE

Additional Editor Comments (optional):

This paper can be accepted for publication now.
---

## [Editor Report · Acceptance letter]

28 Feb 2022

PONE-D-21-27176R1 

Genetic association of ANRIL with susceptibility to Ischemic Stroke: a comprehensive meta-analysis 

Dear Dr. Liu:

I'm pleased to inform you that your manuscript has been deemed suitable for publication in PLOS ONE. Congratulations! Your manuscript is now with our production department. 

Kind regards, 

on behalf of

Dr. Mingqing Xu 

Academic Editor

PLOS ONE